# Pushing the Limits of Gradient Descent for Efficient Learning on Large Images

**Deepak K. Gupta**[*]  *deepak.gupta@iitism.ac.in*
*Transmute AI Lab (Texmin Hub),*
*Indian Institute of Technology (ISM), Dhanbad, India*

**Gowreesh Mago**[*]  *gkrm2727@gmail.com*
*Transmute AI Lab (Texmin Hub),*
*Indian Institute of Technology (ISM), Dhanbad, India*

**Arnav Chavan**[*]  *arnav.chavan@nyunai.com*
*Transmute AI Lab (Texmin Hub),*
*Indian Institute of Technology (ISM), Dhanbad, India*
*Nyun AI, India*

**Dilip Prasad**  *dilip.prasad@uit.no*
*Department of Computer Science,*
*UiT The Arctic University of Norway,*
*Tromso, Norway*

**Rajat Mani Thomas**  *rmt4003@qatar-med.cornell.edu*
*Department of Physiology and Biophysics,*
*Weill Cornell Medicine-Qatar,*
*Qatar-Foundation, Doha, Qatar*

**Reviewed on OpenReview:** *https://openreview.net/forum?id=6dS1jhdemD*

## Abstract

Traditional deep learning models are trained and tested on relatively low-resolution images (< 300 px), and cannot be directly operated on large-scale images due to compute and memory constraints. We propose Patch Gradient Descent (PatchGD), an effective learning strategy that allows us to train the existing CNN and transformer architectures (hereby referred to as deep learning models) on large-scale images in an end-to-end manner. PatchGD is based on the hypothesis that instead of performing gradient-based updates on an entire image at once, it should be possible to achieve a good solution by performing model updates on only small parts of the image at a time, ensuring that the majority of it is covered over the course of iterations. PatchGD thus extensively enjoys better memory and compute efficiency when training models on large-scale images. PatchGD is thoroughly evaluated on PANDA, UltraMNIST, TCGA, and ImageNet datasets with ResNet50, MobileNetV2, ConvNeXtV2, and DeiT models under different memory constraints. Our evaluation clearly shows that PatchGD is much more stable and efficient than the standard gradient-descent method in handling large images, especially when the compute memory is limited. Code is available at **https://github.com/nyunAI/PatchGD**.

## 1 Introduction

In the field of computer vision, deep learning models have emerged as the fundamental framework for advanced feature extraction, significantly outperforming traditional algorithms. For a comprehensive overview of the

---

[*]Equal contribution.

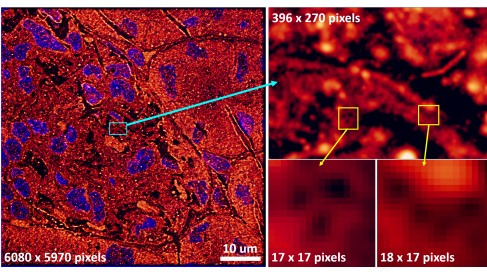

Figure 1: Example nanoscopy image (left) of a mouse kidney cryo-section approximately 1/12th of the area of a single field-of-view of the microscope, chosen to illustrate the level of details at different scales. The bottom right images show that the smallest features in the image of relevance can be as small as a few pixels (here 5-8 pixels for the holes)(Villegas-Hernández et al., 2022).

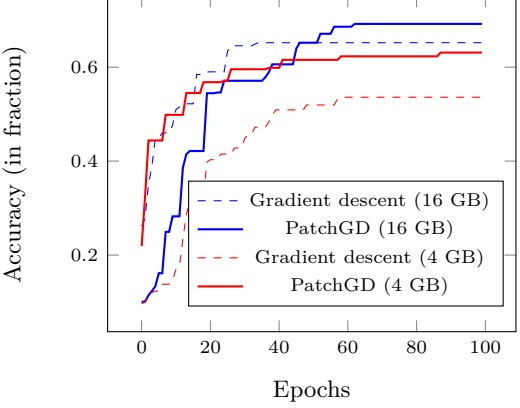

Figure 2: Performance comparison of standard CNN and PatchGD (ours) for the task of classification of UltraMNIST digits of size $512 \times 512$ pixels using ResNet50 model. Two different computational memory budgets of 16 GB and 4GB are used, and it is demonstrated that PatchGD is relatively stable for the chosen image size, even for very low memory compute.

evolution of the associated models and their applications across a wide range of scientific domains, we refer to the reviews presented in Khan et al. (2020); Li et al. (2021a); Alzubaidi et al. (2021); Khan et al. (2022); Shamshad et al. (2022).

With the recent technological developments, very large images are obtained from domains like microscopy (Khater et al., 2020; Schermelleh et al., 2019), medical imaging (Aggarwal et al., 2021), and earth sciences (Huang et al., 2018; Amani et al., 2020), and the challenge of using deep learning models on big data to analyze such images is immense. For example, images obtained from high-content nanoscopy [1] can be as large as $6000 \times 6000$ or even more (Villegas-Hernández et al., 2022), with the smallest scale of features being only a few pixels in size. Clearly, processing these large images with such fine details prohibits the use of any image downsampling algorithm, and using existing models on such high-dimension images is computationally infeasible.

Most prevailing deep learning models are trained on datasets such as ImageNet, which mainly comprise of low-resolution ($< 500$ pixels) images. Most research efforts have focused on achieving state-of-the-art performance on these datasets. However, applying these models to high-resolution images results in a quadratic increase in activation size, requiring significantly more training compute and memory. Moreover, limited GPU memory makes it impractical to process such large images with such models.

This paper introduces a novel training framework for deep learning models aimed at handling very large images. The definition of "large images" depends on the available computational memory for training. For instance, training a ResNet50 model with $10,000 \times 10,000$ size images would be challenging with a 48 GB GPU memory, but training the same model with $512 \times 512$ size images would be feasible with 12 GB GPU memory. However, when limited to a 4 GB GPU memory, even $512 \times 512$ size images may be considered too large.

Figure 2 illustrates the above issue on the task of classification of the UltraMNIST digits (Gupta et al., 2022) into one of the 10 predefined classes labeled from 0-9. More details on the UltraMNIST dataset and classification problem are provided in the supplementary material. The semantic relationship between different parts of the images and the large variation in spatial feature size makes this problem difficult for traditional models, particularly when dealing with large image sizes or low processing memory. In this study,

---

[1]refer Figure 3 in supplementary material

we focus on images of size $512 \times 512$ pixels and examine the problem under two computational memory budgets: GPU memory limits of 4 GB and 16 GB.

For the base model, we use ResNet50 (He et al., 2016) and employ the standard training approach. We refer to it as Gradient descent (GD). Note that the term GD is used here as a generic notation to refer to the class of gradient-based optimizers popularly used in deep learning (such as SGD, SGD with momentum (Bengio et al., 2013) and Adam (Kingma and Ba, 2014), among others), and it is not necessarily restricted to stochastic gradient descent method. For the results demonstrated in Figure 2, we used the Adam optimizer. We also present results using the proposed training method, called Patch Gradient Descent (PatchGD), which is a scalable training approach for building neural networks with large images, low memory compute, or both.

PatchGD's effectiveness is demonstrated in Figure 2, where it outperforms GD in both 16 GB and 4 GB memory limits. The performance difference is 4% at 16 GB but significantly increases to 13% at 4 GB, simulating real-world challenges with large images. Training a ResNet50 model with $512 \times 512$ images using only 4 GB memory leads to inferior performance, as shown in Figure 2. However, PatchGD is stable even at this low memory regime, and this can be attributed to its design which makes it invariant to image size to a large extent. We explain the method in more detail later and present experimental results on various image sizes, highlighting PatchGD's ability to adapt existing CNN models for large images with limited GPU memory.

**Contributions.**

- We present *Patch Gradient Descent (PatchGD)*, a novel strategy to train neural networks on very large images in an end-to-end manner. PatchGD is an adaptation of the conventional feedforward-backpropagation optimization.

- Due to its inherent ability to work with small fractions of a given image, PatchGD is scalable on small GPUs, where training the original full-scale images is not possible.

- PatchGD reinvents the existing training pipeline for deep learning models *to large images* in a very simplified manner and this makes it compatible with any existing CNN architecture or any conventional gradient-based optimization method used in deep learning. Moreover, its simple design allows it to benefit from the pre-training of the standard CNNs on low-resolution data.

## 2 Related Work

This paper seeks to enhance the ability of existing deep-learning models to handle large images. Previous research in this direction is scarce, with most studies focusing on histopathological datasets, which are a popular source of large images. Many of these studies rely on pixel-level segmentation masks, which are not always available. For instance, Iizuka et al. (2020); Liu et al. (2017) use patchwise segmentation masks to perform patch-level classification on whole slide images, and then apply an RNN to obtain the final image label. Meanwhile, Braatz et al. (2022) uses goblet cell segmentation masks for patch-level feature extraction. However, these approaches require labeled segmentation data, are computationally expensive, have limited feature learning, and are more susceptible to error propagation.

Another set of methods focuses on building a compressed latent representation of the large input images using existing pre-trained models or unsupervised learning approaches. For example, Brancati et al. (2021) uses a model pre-trained on Imagenet to construct a latent block, which is further passed to an attention head to do the final classification. Other similar variants include using a U-net autoencoder to build the latent features (Lai et al., 2022), using encoding strategies involving reconstruction error minimization and contrastive learning (Tellez et al., 2018), and getting stronger latent representations through multi-task learning Tellez et al. (2020). A critical limitation of this class of methods is that the encoding network created from unsupervised learning is not always a strong representative of the target task.

There exist several methods that use pre-trained models derived from other tasks as feature extractors and the output is then fed to a classifier. Example methods include using Cancer-Texture Network (CAT-Net) and Google Brain (GB) models as feature extractors (Kosaraju et al., 2022), or additionally using similar

datasets for fine-tuning (Brancati et al., 2021). Although these methods gain an advantage from transfer learning, such two-stage decoupled pipelines propagate errors through under-represented features, and the performance of the model on the target task is hampered. In this paper, we propose a single-step approach that can be trained end-to-end on the target task.

Several studies have aimed to identify appropriate patches from large images and utilize them efficiently for image classification. Naik et al. (2020) suggests constructing a latent space using randomly selected tiles, but this approach fails to maintain semantic coherence across tiles and overlooks features spread across multiple tiles. Campanella et al. (2019) considers this as a multi-instance learning approach, assigning labels to top-K probability patches for classification. Pinckaers et al. (2022); Huang et al. (2022) propose patch-based training but employ streaming convolution networks. Sharma et al. (2021) clusters similar patches and performs cluster-aware sampling for whole-slide image (WSI) and patch classification. Cordonnier et al. (2021) uses a patch scoring mechanism and patch aggregator network for final prediction but involves downsampling for patch scoring, potentially leading to the loss of patch-specific features crucial for WSI. Papadopoulos et al. (2021) progressively increases resolution and localizes regions of interest, disregarding the rest similar to performing hard adaptive attention. DiPalma et al. (2021) trains a high-resolution teacher model and applies knowledge distillation to a lower-resolution version of the same model. Katharopoulos and Fleuret (2019) performs attention sampling on downsampled images and derives an unbiased estimator for gradient updates. However, their attention method involves downsampling, potentially losing vital information. It is important to note that these methods, utilizing patch selection and knowledge distillation, are independent of our work and can be combined with it. However, such integration is beyond the scope of this paper.

With the recent popularity of Transformer-based methods, Chen et al. (2022) proposed a self-supervised learning objective for pre-training large-scale vision transformers at varying scales. Their method involves a vision transformer that leverages the natural hierarchical structure inherent in WSI. However, their method requires a massive pre-training stage which is not always feasible. Also, their method is specific to WSI rather than more general image classification and involves training multiple large-scale transformers. Our method, on the other hand, targets more general image classification tasks and does not involve large-scale pre-training, rather it directly works over any existing CNN model.

To overcome the memory challenges in model training, systems-level methodologies have been explored. Jain et al. (2020); Kirisame et al. (2020) propose approaches like tensor re-materialization to expand available memory, which can be used for both baselines and our PatchGD method, making them complementary. This work primarily focuses on efficiently training large-scale images on low-compute GPUs. Gholami et al. (2018); Dryden et al. (2019); Oyama et al. (2020) address the image size issue by scaling deep learning models across multiple GPUs. There exist other model parallelism methods that can help to handle larger models or activations (Rajbhandari et al., 2020; Shoeybi et al., 2019). These approaches distribute a model across multiple GPUs, effectively managing memory issues by leveraging the collective memory capacity. However, PatchGD is designed to help models operate under low memory settings and it helps to push the limits of each single GPU beyond its conventional capacity. Thus, PatchGD and model parallelism are orthogonal techniques that can be combined to further enhance the performance, allowing for the distribution of large models across multiple GPUs while maximizing the utilization of each GPU's capacity through PatchGD.

Two approaches in the current literature enable training a larger model on a low-memory GPU, activation checkpointing (Chen et al., 2016) and activation offloading (Rhu et al., 2016; Cui et al., 2016). Checkpointing saves only the necessary activations during forward propagation and recomputes the remaining from the saved ones as needed during backpropagation. Offloading shifts part of the activations to the CPU when the GPU memory starts to approach its full capacity and retrieves them when needed. While both these approaches have been effective, they lead to a significant increase in latency in the learning process. More importantly, the gain in batch size during training is significantly lower when compared to PatchGD. We shed light on both these aspects through numerical experiments later in this paper.

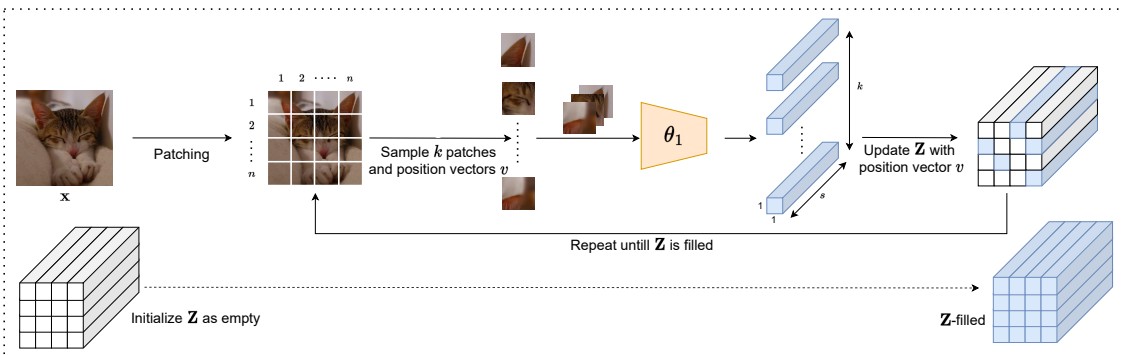

(a) Pipeline for the filling of **Z** block, also referred as **Z**-filling.

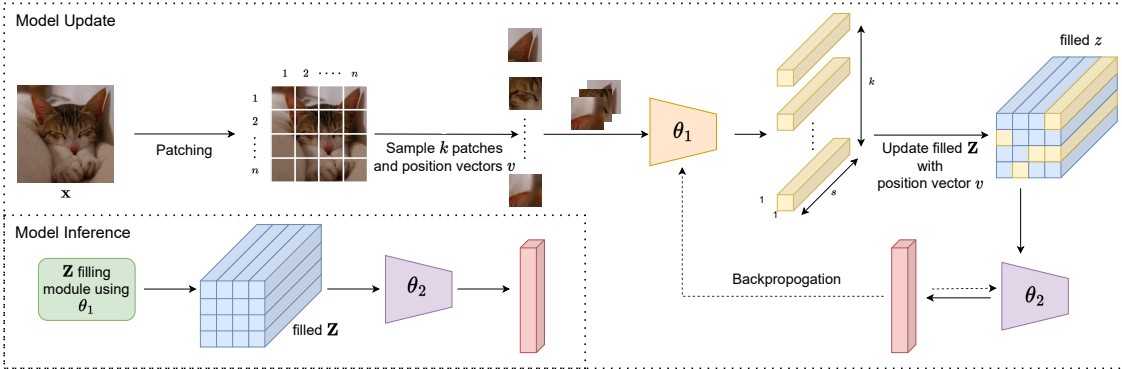

(b) Model update and model inference.

Figure 3: Schematic representations of the pipelines demonstrating working of different components of the PatchGD process.

## 3 Approach

### 3.1 General description

*Patch Gradient Descent (PatchGD)* is a novel approach for training deep learning models with high-resolution images. It's a modification of the standard feedforward-backpropagation method. PatchGD is built on the hypothesis that, instead of applying gradient-based updates to the entire image simultaneously, similar results can be achieved by updating the model in small image segments, while ensuring the full image coverage over multiple iterations. Even if only a portion of the image is used in each iteration for gradient updates, the model is still trainable end-to-end with PatchGD.

Figure 3 presents the schematic representation of the PatchGD pipeline. The central idea behind PatchGD is to construct the **Z** block, which is a deep latent representation of the entire input image. Although only a subset of the input is used to perform model updates, **Z** captures information about the entire image by combining information from different parts of the image acquired from the previous update steps. Figure 3a illustrates the use of the **Z** block, which is an encoding of an input image **X** using a model parameterized by weights $\boldsymbol{\theta}_1$. The input image is divided into $n \times n$ number of patches, and each patch is processed independently using $\boldsymbol{\theta}_1$. The size of **Z** is always enforced to be $n \times n \times s$, such that each patch in the input space corresponds to the respective $1 \times 1 \times s$ segment in the **Z** block.

The filling of **Z** is carried out in multiple steps, with each step involving the sampling of $k$ patches and their positions from **X** and feeding them to the model as a batch for processing. The output from the model along with the corresponding positions are then used to fill the respective parts of **Z**. After sampling all $n \times n$ patches of **X**, the completely filled **Z** is obtained. PatchGD utilizes this concept of **Z**-filling during both training and inference stages.

To create an end-to-end model, we incorporate a small subnetwork that consists of convolutional/self-attention and fully-connected layers. This subnetwork processes the information contained in $\mathbf{Z}$ and converts it into a $c$-dimensional classification probabilities. It is worth noting that the computational cost of adding this small subnetwork is minimal. The Figure 3b illustrates the pipelines for both model training and inference stages. During training, the model components $\boldsymbol{\theta}_1$ and $\boldsymbol{\theta}_2$ are updated. At every model update step, only a fraction of the patches are updated in $\mathbf{Z}$ while the rest is retained from the previous state of the model. Subsequently, we use the partially updated $\mathbf{Z}$ to calculate the loss function value and update the model parameters using backpropagation.

**Why does PatchGD work?** PatchGD operates on the principle that patches remain independent until the $\mathbf{Z}$ block, with no interactions required prior to this stage. Unlike conventional methods, information exchange at patch boundaries is deferred. Nevertheless, the negligible loss incurred as a result of this delayed interaction is minimal in comparison to the patch size. This effect diminishes further when applied to large images and correspondingly large patches.

## 3.2 Mathematical formulation

In this section, we present a detailed mathematical formulation of the proposed PatchGD approach. For simplicity, we describe it in the context of image classification.

Let $f_{\boldsymbol{\theta}} : \mathbb{R}^{M \times M \times C} \to \mathbb{R}^c$ denote a model parameterized by $\boldsymbol{\theta}$ that takes an input image $\mathbf{X}$ of spatial size $M \times M$ and $C$ channels and computes the probability of it to belong to each of the $c$ pre-defined classes. To train this model, the following optimization problem is solved.

$$\min_{\boldsymbol{\theta}} \ \mathcal{L}(f(\boldsymbol{\theta}; \mathbf{X}), \mathbf{y}), \tag{1}$$

where $\mathbf{X}, \mathbf{y} \in \mathcal{D}$ represents the data samples used, and $\mathcal{L}(\cdot)$ represents the loss function. Conventionally, this problem is solved using mini-batch gradient descent method where at every step, the model weights are updated using the average of gradients computed over a batch of samples, denoted here as $\mathcal{S}$. Based on this, the model update at the $i^{\text{th}}$ step is

$$\boldsymbol{\theta}^{(i)} = \boldsymbol{\theta}^{(i-1)} - \frac{\alpha}{B} \sum_{\mathbf{X} \in \mathcal{S}} \frac{\mathrm{d}\mathcal{L}^{(\mathbf{X})}}{\mathrm{d}\boldsymbol{\theta}^{(i-1)}} \tag{2}$$

where $\alpha$ and $B$ denote the learning rate and the size of the batch used, respectively. As can be seen in Eq. 2, if the size of image samples $s \in \mathcal{S}$ is very large, it will lead to large memory requirements for the respective activations, and under limited compute availability, only small values of $B$, and sometimes not even a single sample, fit into the GPU memory. This should clearly demonstrate the limitation of the gradient descent method when handling large images. This issue is alleviated by our PatchGD approach.

---

**Algorithm 2** Filling of the $\mathbf{Z}$ block (referred as $\mathbf{Z}$-filling)

---

**Input:** Input image $\mathbf{X} \in \mathbb{R}^{M \times M \times C}$, Pre-trained feature extractor $f_{\boldsymbol{\theta}_1}$, Patch size $p$, $m \leftarrow (M/p)$
**Initialize:** $\mathbf{Z} \in \mathbb{R}^{m \times m \times s}$, `requires_gradient`$(f_{\boldsymbol{\theta}_1}) = False$
**repeat**
  $\mathbf{x}_{a,b} \leftarrow$ `patch_extractor`$(\mathbf{X}, a, b)$                               *#Extract patch at location (a,b)*
  $\mathbf{x}_{a,b} \in \mathbb{R}^{p \times p \times C}$
  $\mathbf{z}_{a,b} \leftarrow f_{\boldsymbol{\theta}_1}(\mathbf{x}_{a,b}), \mathbf{z}_i \in \mathbb{R}^{1 \times 1 \times s}$                *#Extract embedding from each patch*
  $\mathbf{Z}[a,b] \leftarrow \mathbf{z}_{a,b}$                                 *#Update the positional embedding*
**until** all patches sampled
**Return $\mathbf{Z}$ =0**

---

**PatchGD.** As described in Section 3.1, PatchGD avoids model updates on an entire image sample in one go, rather it computes gradients using only part of the image and updates the model parameters. In this regard,

---

**Algorithm 1** Model Training for 1 iteration

---

1: **Input:** Batch of input images $\mathcal{X} \in \mathbb{R}^{B \times M \times M \times C}$, Pre-trained feature extractor $f_{\boldsymbol{\theta}_1}$, Classifier head $g_{\boldsymbol{\theta}_2}$, Patch size $p$, Inner iterations $\zeta$, Patches per inner iteration $k$, Batch size $B$, Learning rate $\alpha$, Grad. Acc. steps $\epsilon$          $\#M^2 \div p^2 = \zeta \times k$
2: **Initialize:** $\mathbf{Z} \leftarrow \mathbf{0}^{B \times m \times m \times c}, \mathbf{U}_1 \leftarrow \mathbf{0}, \mathbf{U}_2 \leftarrow \mathbf{0}$
3: $\mathbf{Z} \leftarrow \mathbf{Z}\text{-filling}(\mathbf{X}, f_{\boldsymbol{\theta}_1}, p)$ forall $\mathbf{X} \in \mathcal{X}$
4: `requires_gradient`$(f_{\boldsymbol{\theta}_1}) = True$
5: **for** $j : 1$ to $\zeta$ **do**
6:      **for** $\mathbf{X}$ in $\mathcal{X}$ **do**
7:          $\{\mathcal{P}_{\mathbf{X},j}, v\} \leftarrow$ `patch_sampler`$(\mathbf{X}, k)$,                 *#Sample k patches from each image*
8:          $\mathcal{P}_{\mathbf{X},j} \in \mathbb{R}^{p \times p \times C \times k}$
9:          $\mathbf{z} \leftarrow f_{\boldsymbol{\theta}_1}(\mathcal{P}_{\mathbf{X},j})$                 *#Extract embedding from each patch*
10:          $\mathbf{Z}[v] \leftarrow \mathbf{z}$                 *#Update the positional embeddings*
11:          $\mathbf{y}_{\text{pred}} \leftarrow g_{\boldsymbol{\theta}_2}(\mathbf{Z})$                 *# Classifier prediction over updated $\mathbf{Z}$*
12:          $\mathcal{L} \leftarrow$ `calculate_loss`$(\mathbf{y}, \mathbf{y}_{\text{pred}})$
13:          $\mathbf{U}_1 \leftarrow \mathbf{U}_1 + \mathrm{d}\mathcal{L}/\mathrm{d}\boldsymbol{\theta}_1, \mathbf{U}_2 \leftarrow \mathbf{U}_2 + \mathrm{d}\mathcal{L}/\mathrm{d}\boldsymbol{\theta}_2$       *#Accumulate gradients across batch of patches*
14:      **end for**
15:      **if** $j\%\epsilon = 0$ **then**
16:          $\mathbf{U}_1 \leftarrow \mathbf{U}_1/\epsilon, \mathbf{U}_2 \leftarrow \mathbf{U}_2/\epsilon$
17:          $\boldsymbol{\theta}_1 \leftarrow \boldsymbol{\theta}_1 - \alpha\mathbf{U}_1, \boldsymbol{\theta}_2 \leftarrow \boldsymbol{\theta}_2 - \alpha\mathbf{U}_2$           *#Update weights every $\epsilon$ inner iterations*
18:          $\mathbf{U}_1 \leftarrow \mathbf{0}, \mathbf{U}_2 \leftarrow \mathbf{0}$
19:      **end if**
20: **end for**$=0$

---

the model update step of PatchGD can be stated as

$$\boldsymbol{\theta}^{(i,j)} = \boldsymbol{\theta}^{(i,j-1)} - \frac{\alpha}{k \cdot B_i} \sum_{\mathbf{X} \in \mathcal{S}_i} \sum_{p \in \mathcal{P}_{\mathbf{X},j}} \frac{\mathrm{d}\mathcal{L}^{(\mathbf{X},p)}}{\mathrm{d}\boldsymbol{\theta}^{(i,j-1)}}. \tag{3}$$

Here, $i$ refers to a mini-batch iteration within a certain epoch. Further, $j$ denotes the inner iterations, where at every inner iteration, $k$ patches are sampled from the each input image $X \in \mathbf{X}$ (denoted as $\mathcal{P}_{\mathbf{X},j}$) and the gradient-based updates are performed as stated in Eq. 3. Note that for any iteration $i$, multiple inner iterations are run ensuring that the majority of samples from the full set of patches that are obtained from the tiling of $\mathbf{X}$ are explored.

In Eq. 3, $\boldsymbol{\theta}^{(i,0)}$ denotes the initial model for the inner iterations on $\mathcal{S}_i$ and is equal to $\boldsymbol{\theta}^{(i-1,\zeta)}$, the final model state after $\zeta$ inner iterations of patch-level updates using $\mathcal{S}_{i-1}$. For a more detailed understanding of the model update process, please see Algorithm 1. As described earlier, PatchGD uses an additional sub-network that looks at the full latent encoding $\mathbf{Z}$ for the input batch $\mathbf{X}$. Thus, the parameter set $\boldsymbol{\theta}$ is extended as $\boldsymbol{\theta} = [\boldsymbol{\theta}_1, \boldsymbol{\theta}_2]^\intercal$, where the base CNN model and the additional sub-network are $f_{\boldsymbol{\theta}_1}$ and $g_{\boldsymbol{\theta}_2}$, respectively.

Algorithm 1 describes model training over one batch of $B$ images, denoted as $\mathbf{X} \in \mathbb{R}^{B \times M \times M \times C}$. As the first step of the model training process, $\mathbf{Z}$ corresponding to $\mathbf{X}$ is initialized. The process of filling of $\mathbf{Z}$ is described in Algorithm 2. For a patch indexed by position $v$ and denoted as $\mathbf{x}_v$, the respective $\mathbf{Z}[v]$ is updated using the output obtained from $f_{\boldsymbol{\theta}_1}$. Note here that $\boldsymbol{\theta}_1$ is loaded from the last state obtained during the model update on the previous batch of images. During the filling of $\mathbf{Z}$, no gradients are stored for backpropagation.

Next, the model update process is performed over a series of $\zeta$ inner-iterations, where at every step $j \in \{1, 2, \ldots, \zeta\}$, $k$ patches are sampled per image in $\mathbf{X}$ and the respective parts of $\mathbf{Z}$ are updated. Next, the partly updated $\mathbf{Z}$ is processed with the additional sub-network $\boldsymbol{\theta}_2$ to compute the class probabilities and the corresponding loss value. Based on the computed loss, gradients are backpropagated to perform updates of $\boldsymbol{\theta}_1$ and $\boldsymbol{\theta}_2$. Note that we control here the frequency of model updates in the inner iterations through an additional term $\epsilon$. Similar to how a batch size of 1 in mini-batch gradient descent introduces noise and adversely affects the convergence process, we observed that gradient update per inner iteration leads to sometimes poor convergence. Thus, we introduce gradient accumulation over $\epsilon$ inner steps and

Table 1: Performance scores on the UltraMNIST dataset with images of size $512 \times 512$ obtained using ResNet50 architecture. Patch size for PatchGD is 256.

| Method | Batch size | Memory (GB) | Acc. % |
|---|---|---|---|
| GD | 27 | 16 | 65.2 |
| GD*[3] | 26 | 16 | 50.5 |
| PatchGD | 100 | 16 | **69.2** |
| GD | 2 | 4 | 53.6 |
| GD*[3] | 2 | 4 | 52.5 |
| PatchGD | 7 | 4 | **63.1** |

Table 2: Performance scores on the UltraMNIST dataset with $512 \times 512$ images obtained using MobileNetV2 architecture. Patch size for PatchGD is 256.

| Method | Batch size | Memory (GB) | Acc. % |
|---|---|---|---|
| GD | 30 | 16 | 67.3 |
| GD*[3] | 30 | 16 | 64.3 |
| PatchGD | 120 | 16 | **83.7** |
| GD | 3 | 4 | 67.7 |
| GD* [3] | 3 | 4 | 60.0 |
| PatchGD | 10 | 4 | **74.8** |

update the model accordingly. Gradients are allowed to backpropagate only through those parts of $\mathbf{Z}$ that are active at the $j^{\text{th}}$ inner-iteration. During inference phase, $\mathbf{Z}$ is filled using the optimized $f_{\boldsymbol{\theta}_1^*}$ as described in Algorithm 2, and then the filled version of $\mathbf{Z}$ is used to compute the class probabilities for input $\mathbf{X}$ using $g_{\boldsymbol{\theta}_2^*}$.

## 4 Experiments

We showcase the effectiveness of PatchGD on two benchmark datasets with large images and multiple scales, and ablation studies on multiple models, datasets and use cases.

### 4.1 Experimental setup

**Datasets.** We perform thorough evaluation on two datasets, UltraMNIST (Gupta et al., 2022) and PANDA (Bulten et al., 2022), and also conduct additional experiments using TCGA-NSCLC (Chen et al., 2022) and ImageNet Deng et al. (2009) datasets. Details about the datasets are presented in the supplementary part of the paper.

**CNN models.** We assess PatchGD on ResNet50 and MobileNetV2 architectures. ResNet50 serves as a backbone for diverse computer vision tasks, while MobileNetV2 is a lightweight architecture for edge devices. We also conduct experiments with ConvNextV2, a state-of-the-art vision model, as well as provide preliminary results for generative modeling.

**Implementation details.** We employ consistent hyperparameters throughout our experiments and report classification accuracy for UltraMNIST and ImageNet tasks and additionally Quadratic Weighted Kappa (QWK)[2] on the PANDA dataset. For TCGA-NSCLC, we comply with the previous baselines and report the mean and standard deviation of AUC across a 10-fold cross-validation set. Both the baselines and PatchGD are implemented using PyTorch. We consider GPU memory constraints of 4GB, 16GB, and 48GB to simulate common limits and measure latency on an NVIDIA 40GB A100 GPU and an NVIDIA 24GB L4 GPU. Additional details are described in the supplementary material.

### 4.2 Results

**UltraMNIST classification.** The performance of PatchGD for UltraMNIST has already been shown in Figure 2. More detailed results are presented in Tables 1 and 2. For both the architectures, PatchGD improves over the standard gradient descent method (abbreviated as GD) by large margins. Our approach employs an additional sub-network $g_{\boldsymbol{\theta}_2}$, and it can be argued that the gains reported in the paper are due to it. For this purpose, we extend the base CNN architectures used in GD and report the respective performance scores in Tables 1 and 2 as GD*. [3].

For both architectures, PatchGD outperforms GD as well as GD* by large margins. For ResNet50, the performance difference is even higher for a low memory constraint. At 4 GB, while GD seems unstable with

---

[2]QWK: metric on histopathological images and PANDA dataset.
[3]GD* refers to baseline being extended with the same sub-network $g_{\boldsymbol{\theta}_2}$.

Table 3: Performance scores obtained using Resnet50 on PANDA dataset for Gradient Descent (GD) and Patch Gradient Descent (PatchGD).

| Method | Resolution | Patch Size | Batch Size | Mem. (GB) | Throughput (imgs/sec) | Accuracy % | QWK |
|---|---|---|---|---|---|---|---|
| Baseline | 512 | - | 27 | 16 | 618.05 | 44.4 | 0.558 |
| PatchGD | 512 | 128 | 86 | 16 | 521.42 | 44.9 | 0.576 |
| PatchGD | 512 | 64 | 200 | 16 | 341.87 | **52.1** | **0.616** |
| Baseline | 2048 | - | 1 | 16 | 39.04 | 34.8 | 0.452 |
| PatchGD | 2048 | 128 | 14 | 16 | 32.52 | 53.9 | 0.627 |
| Baseline | 2048 | - | 6 | 48 | 39.04 | 49.4 | 0.625 |
| PatchGD | 2048 | 128 | 56 | 48 | 32.52 | **56.2** | **0.667** |
| Baseline | 4096 | - | 1 | 48 | 9.23 | 50.0 | 0.611 |
| PatchGD | 4096 | 256 | 26 | 48 | 9.62 | **59.7** | **0.730** |

Table 4: Comparison of PatchGD (ResNet50 architecture) with existing methods at 4096 image size and 48GB memory constraint.

| Method | Accuracy % | QWK |
|---|---|---|
| HIPT Chen et al. (2022) | 34.8 | 0.388 |
| HIPT-L | 49.3 | 0.531 |
| ABNN Brancati et al. (2021) | 48.2 | 0.593 |
| C2C Sharma et al. (2021) | 50.9 | 0.668 |
| PatchGD | **59.7** | **0.730** |

a performance dip of more than 11% compared to the 16 GB case, PatchGD is significantly more stable. For MobileNetV2, the difference between PatchGD and GD is even higher at 16GB case, thereby clearly showing that PatchGD blends well with even lightweight models such as MobileNetV2. For MobileNetV2, there is no drop in model performance when going from 16 GB to 4 GB, which demonstrates that MobileNetV2 can work well with GD even at low memory conditions. Nevertheless, PatchGD still performs significantly better. The underlying reason for this gain can partly be attributed to the fact that since PatchGD facilitates operating with partial images, the activations are small and more images per batch are permitted. We also observe that the performance scores of GD* are inferior compared to even GD. ResNet50 and MobilenetV2 are optimized architectures and we speculate that the addition of plain convolutional layers in the head of the network is not suited due to which the overall performance is adversely affected.

**Prostate Cancer Classification (PANDA).** Table 3 presents the results obtained on PANDA dataset for three different image resolutions. For all experiments, we maximize the number of images used per batch while also ensuring that the memory constraint is not violated. For images of $512 \times 512$, we see that PatchGD, with patches of size $128 \times 128$, delivers approximately the same performance as GD (for both accuracy as well as Quadratic Weighted Kappa (QWK) metric at 16 GB memory limit. However, reducing the patch size and thus increasing the batch size leads to a very sharp gain in the scores of PatchGD. For a similar memory constraint, when images of size $2048 \times 2048$ pixels are used, GD scores approximately 10% lower while PatchGD shows a boost of 9% in accuracy.

Two factors contribute to the performance gap between GD and PatchGD. Firstly, GD faces a bottleneck with batch size due to increased activation size in higher-resolution images, allowing only 1 image per batch. Gradient accumulation and hierarchical training were explored but did not improve performance significantly. Increasing the memory limit helped mitigate the issue of using only 1 image per batch. Secondly, the optimized receptive field of ResNet50 is not well-suited for higher-resolution images, resulting in suboptimal performance. PatchGD demonstrates superior accuracy and QWK compared to GD on the PANDA dataset when handling large images end-to-end. In terms of inference latency, PatchGD performs comparably to GD. The smaller activations in PatchGD offset the slowness caused by patchwise image processing. PatchGD shows potential for real-time inference in applications requiring large image handling.

We further present a comparison of PatchGD with the existing methods designed for handling large images, and the results are presented in Table 4. We used Resnet50 at 4096 image resolution and a 48GB GPU

memory constraint for training. Note that almost all works that exist on handling large images are not designed to work with memory constraints, and if put in such applications, these lead to unstable performance scores. For example, although the vision transformer backbones of HIPT are pre-trained on large medical datasets, the performance of the model in the memory-constrained setting is lowest among the 4 methods presented in the table. For HIPT, all the layers of the vision transformer backbones are trainable, and a batch size of only 5 fits in the memory. This is the primary reason for the significant drop in the performance of the method. The original HIPT model is trained with large batch sizes over a set of GPUs, however, in our memory-constrained setup, it is not possible. The performance of ABNN and C2C is relatively better, however, they are still significantly lower than the PatchGD training of a simple architecture. C2C employs attention modules in the head of the network, and we believe with such additions, the performance of PatchGD could be boosted even further. Nevertheless, we see from the presented results that for memory-constrained settings, PatchGD performs significantly better than any other existing method when it comes to handling large images.

Since HIPT uses a transformer model, one possible way to enhance its performance under low memory setting is to use layer normalization and implement gradient accumulation over a series of iterations. We conducted an experiment with gradient accumulation over 12 steps, referred to as HIPT-L in Table 4. This led to an equivalent batch size of 60. Although the convergence was slow, the performance of the model boosted from 34.8 to 49.3. This clearly demonstrates that transformers with gradient accumulation could work well even at low batch sizes. Nevertheless, we still see a significant performance gap of more than 10% between HIPT and our approach. Moreover, transformers are known to be data-hungry and one important thing to note here is that the pre-trained HIPT model we are using in this paper is already heavily trained on a very large medical dataset comprising training images from a variety of medical datasets. On the contrary, our model is only pre-trained on standard ImageNet and no additional pre-training is done. This clearly makes our approach stand out when compared to HIPT in the sense that it is applicable for low memory as well as relatively low training data regimes as well.

**TCGA-NSCLC classification with no pre-training.** To further demonstrate the efficacy of PatchGD on established large image benchmarks, we study the task of TCGA-NSCLC dataset classification and compare our solution with popular approaches such as HIPT (Chen et al., 2022) and CLAM-SB Lu et al. (2021), among others. Related results are presented in Table 5. Note here that the HIPT model uses a 3-stage Transformer model with a ViT backbone which is pre-trained on an external large-scale histopathological dataset first and then fine-tuned on TCGA-NSCLC data. Further, it uses the images at a gigapixel scale. Further, CLAM-SB uses a multistage processing approach where a segmentation map is first obtained, followed by creating embedding of small patches. Attention pooling is then used to assign weight to each patch which together are then served to a classification model. The other baselines listed in Table 5 similarly also use multistage processing with additional pretraining done on external datasets for boosted discriminative power.

For our study on PatchGD, we employed a lightweight ConvNeXt-Tiny model without any pretraining on external histopathological datasets. Initially, we established two baselines (Baseline-1 and Baseline-2) by training the model using standard backpropagation techniques. We then applied the PatchGD methods for training. Despite using only 4K images, no external pretraining, and a simple tiny backbone, PatchGD outperformed HIPT and other methods designed for handling large images, demonstrating its superiority. Notably, Baseline-2 also outperformed GCN-MIL and MIL, highlighting the effectiveness of the ConvNeXt architecture and the simplicity of the training process. We believe that incorporating pre-training on external data, as done in HIPT, could further enhance the performance of the PatchGD solution.

**PatchGD with transformer architecture.** The simplicity of our PatchGD approach allows its easy integration with transformer architecture as well. We demonstrate its compatibility with transformer architectures through experiments on the ImageNet dataset using DeiT-Tiny architecture. Results related to these are presented in Table 6. We observe that PatchGD couples well with the chosen model and leads to improved performance over the respective baselines obtained using conventional training. Note that the baseline is optimized using the high-performing training procedure described in Touvron et al. (2021) and the performance is further optimized across 3 different image resolutions.

Table 5: Performance comparison of PatchGD with HIPT and other baseline models for the task of TCGA-NSCLC subtyping dataset.

| Method | Model | Image Size | AUC | Standard Deviation |
|--------|-------|-----------|-----|-------------------|
| Baseline-1 | ConvNeXt-V2 Tiny | 224 | 78.0 | 3.7 |
| GCN-MIL (Zhao et al., 2020) | VAE-GAN + Graph CNN | | 83.1 | 3.4 |
| MIL (Lu et al., 2021) | | | 89.2 | 4.2 |
| Baseline-2 | ConvNeXt-V2 Tiny | 4096 | 90.4 | 4.3 |
| DS-MIL (Li et al., 2021b) | Patching + Resnet18 + Aggregation | | 92.0 | 2.4 |
| CLAM-SB (Lu et al., 2021) | Patching + Resnet50 + Attention Clustering | | 92.8 | 2.1 |
| HIPT (Chen et al., 2022) | Patching + 3 × Transformer | - | 95.2 | 2.1 |
| PatchGD | ConvNeXt-V2 Tiny | 4096 | **97.0** | **1.7** |

Table 6: Accuracy Comparison for ImageNet on Deit-Tiny architecture with Gradient Descent and PatchGD.

| # Classes | # Samples / Class | Baseline Accuracy (%) | | | PatchGD Accuracy (%) |
|-----------|-------------------|------|------|------|------|
| | | 224 | 384 | 512 | 512 |
| 25 | 100 | 85.76 | 88.68 | 88.72 | 90.74 |
| 25 | 200 | 88.41 | 90.32 | 90.16 | 92.12 |
| 25 | 500 | 90.08 | 92.00 | 92.18 | 93.14 |
| 25 | 1000 | 91.28 | 93.12 | 93.20 | 95.44 |
| 10 | 100 | - | 85.40 | - | 88.40 |
| 25 | 100 | - | 88.68 | - | 90.74 |
| 100 | 100 | - | 76.90 | - | 78.20 |
| 500 | 100 | - | 73.65 | - | 70.82 |

For transformer backbones, we have observed that the performance of the model is better when the head is also a transformer rather than a CNN model. Table 7 presents a comparison of CNN and transformer heads for the classification task on the PANDA dataset. For the transformer head, we use a single multi-headed self-attention layer with 2 heads each of 192 channels followed by a linear layer. The CNN head uses 3 conv-relu-bn blocks with a kernel size of $3 \times 3$ and 256 channels followed by a linear layer. We consistently see that the transformer head works better.

**Handling natural images (ImageNet).** To understand how PatchGD works with natural images, we study its performance on ImageNet dataset for different choices of the number of classes as well as the number of samples per class. This follows the results discussed in the earlier section. We conduct these experiments using DeiT-Tiny transformer architecture and the results are reported in Table 6. To study the effect of the number of samples, we fix classes to 25. Interestingly, we observe that PatchGD outperforms the standard training approach by around 2% accuracy, a significant improvement in the context of ImageNet training.

We further examined the performance of PatchGD across different numbers of classes, keeping the number of samples per class fixed at 100 (Table 6). Interestingly, PatchGD outperformed the baseline approach when dealing with fewer classes. However, when the number of classes increased to 500, the baseline method performed better. This discrepancy arises because, for low-resolution images such as those in the ImageNet dataset, the small information loss at the edges of the patches becomes significant when there are many classes and limited samples per class. Our initial findings suggest that this issue can be mitigated to some extent by using overlapping patches, although this increases computational demands. Nonetheless, our observations indicate that PatchGD is the preferred choice for natural images in low-data regimes.

**PatchGD vs. Activation Checkpointing vs. Activation Offloading.** As has been described throughout this paper, PatchGD aims at better utilization of GPUs, by facilitating to training of deep learning models with larger images (leading to larger model activations) on smaller GPUs. Two other popular approaches aiming at fitting a larger model on smaller GPUs are activation checkpointing (Chen et al., 2016) and

Table 7: Performance scores obtained using DeiT-Small on PANDA dataset for Gradient Descent (GD) and Patch Gradient Descent (PatchGD).

| Method | Resolution | Patch Size | Batch Size | Head | Mem. (GB) | Accuracy % | QWK |
|--------|-----------|-----------|-----------|------|-----------|-----------|-----|
| Baseline | 512 | - | 22 | - | 16 | 48.4 | 0.596 |
| PatchGD | 512 | 128 | 136 | CNN | 16 | 44.9 | 0.576 |
| PatchGD | 512 | 128 | 136 | Transformer | 16 | 48.7 | 0.599 |
| Baseline | 2048 | - | 4 | - | 48 | 48.6 | 0.612 |
| PatchGD | 2048 | 128 | 32 | CNN | 48 | 48.9 | 0.589 |
| PatchGD | 2048 | 128 | 32 | Transformer | 48 | 57.4 | 0.702 |

activation offloading (Rhu et al., 2016). Table 6 presents a comparison of PatchGD with these methods. We present the comparison of ResNet50 architecture on PANDA dataset at two different image resolutions on an NVIDIA 16 GB L4 graphics card. For gradient checkpointing, we employ chunk sizes of 4 and 6.

PatchGD outperforms checkpointing and offloading approaches, particularly with 2K resolution images, where the margin of superiority is significantly larger. Under the selected memory constraint at this resolution, both baseline methods can only handle a maximum batch size of 4 per iteration, with activation offloading managing only 2. In contrast, PatchGD can handle batch sizes of 14. For smaller images, all methods can increase the batch size, but PatchGD still delivers the best performance. This clearly demonstrates that PatchGD is more effective in utilizing GPU resources. Additionally, it is worth noting that PatchGD, checkpointing, and offloading are orthogonal methods and can be combined to fit even larger models on smaller GPU resources.

Table 8: Performance comparison of PatchGD against Activation Offloading (Rhu et al., 2016) and Activation Checkpointing (Chen et al., 2016) on PANDA dataset using ResNet50 model and a NVIDIA L4 16 GB GPU.

| Method | Image Size | Batch Size | Peak Memory (GB) | Training time / image / iteration(ms) | Accuracy |
|--------|-----------|-----------|-----------------|--------------------------------------|----------|
| Baseline | 2048 | 1 | 8.6 | 716 | 34.8 |
| Activation Off-loading | 2048 | 2 | 14.6 | 2603 | 42.1 |
| Gradient Checkpointing, chunks=4 | 2048 | 3 | 14.1 | 1007 | 46.0 |
| Gradient Checkpointing, chunks=6 | 2048 | 4 | 13.1 | 998 | 48.0 |
| PatchGD | 2048 | 14 | 15.1 | 930 | 56.2 |
| Baseline | 512 | 27 | 14.6 | 42 | 44.4 |
| Activation Off-loading | 512 | 32 | 14.9 | 161 | 46.8 |
| Gradient Checkpointing, chunks=4 | 512 | 52 | 14.8 | 59 | 46.2 |
| Gradient Checkpointing, chunks=6 | 512 | 72 | 14.6 | 59 | 44.7 |
| PatchGD | 512 | 200 | 14.7 | 79 | 52.1 |

**Additional study.** In this section, we show some additional experiments to further prove the advantages of PatchGD. Training recipes and hyperparameters are provided in the supplementary material.

*Role of end-to-end training.* Table 9 shows that freezing the backbone leads to reduced performance, highlighting the key role of end-to-end training in PatchGD. Other existing methods can fine-tune the network

Table 9: Performance scores on PANDA dataset (2048 × 2048) at 24 GB memory budget for different choices of the feature extractor: pretrained and frozen on ImageNet, trained on PANDA using GD and frozen, and fully trainable.

| Feature extractor | Batch Size | QWK | Accuracy% |
|-------------------|-----------|-----|-----------|
| ImageNet (frozen) | 50 | 0.538 | 44.6 |
| PANDA (frozen) | 50 | 0.642 | 50.0 |
| Fully-Trainable | 40 | 0.662 | 56.0 |

Table 10: Performance scores for ConvNextV2 on PANDA dataset.

| Method | Image Size | Batch Size | Mem. (GB) | Acc.% |
|--------|-----------|-----------|-----------|-------|
| PatchGD | 2048 | 40 | 24 | 49.3 |
| Baseline | 2048 | 4 | 24 | 45.1 |
| PatchGD | 512 | 128 | 16 | 44.0 |
| Baseline | 512 | 50 | 16 | 43.9 |

end-to-end, but only on low-resolution images, whereas PatchGD enjoys fully-trainable end-to-end training even at higher resolutions.

*Additional architectures.* Beyond the experiment on TCGA-NSCLC task, we also conducted an additional experiment with ConvNext-V2, a state-of-the-art image classification model, on PANDA dataset and the results are presented in Table 10). PatchGD outperforms the baseline at higher resolution (2048) while performing competitively at a relatively low resolution too (512). This shows that PatchGD can take advantage of the higher representation power of newer CNN architectures.

*On attention-based head module for CNNs.* We have shown in the paper the working of a CNN backbone with a CNN head as well as a Transformer backbone with a Transformer head. Here, we study whether using a head with attention module could be beneficial for the learning of CNNs. For this purpose, we experimented with ResNet50 backbone and PatchGD and we replaced the CNN head with an attention-based MLP to see the effect. For the attention head, we employed a single-layer multi-head attention module with 64 heads, and for each pixel of the latent corresponding to a token, we concatenated a trainable CLS token for final classification. Additional details are presented in the supplementary material. Compared to the base performance accuracy of 52.1% with a CNN head, the accuracy of the model with the attention head improved to 53.6%. This clearly shows that using an attention module can help to enhance the performance of PatchGD results. Further, we anticipate that for larger images, where the spatial size of the L1-block is larger, this improvement will be even more.

*Hyperparameter study.* Table 11 shows the impact of patch sampling on PatchGD's performance. We find that using smaller sampling fractions per inner iteration leads to improved accuracy, which is counter-intuitive since smaller fractions provide less image context. This behavior may be attributed to regularization noise induced by smaller patch batch sizes, benefiting the convergence process. However, further research is needed for a comprehensive understanding. Additionally, the fraction of the image seen in one pass does not significantly affect performance except when it is low, as insufficient context impedes convergence.

Table 11: Sampling ablation on PANDA dataset. Memory limit is 16 GB, Image size and patch size are 2048 and 128 respectively.

| Sampling | Max Sampled | Accuracy % | QWK |
|---|---|---|---|
| 50 | 100 | 42.3 | 0.538 |
| 30 | 100 | 49.9 | 0.613 |
| 10 | 100 | 53.9 | 0.627 |
| 10 | 70 | 53.1 | 0.624 |
| 10 | 50 | 53.9 | 0.622 |
| 10 | 30 | 51.1 | 0.610 |

We also explore the influence of the gradient accumulation length parameter on PatchGD, and the results are available in the supplementary material. We observe that performing gradient-based updates per inner iteration yields superior performance in our experiment. However, the choice of $\epsilon$ depends on the number of inner steps $\zeta$. For large $\zeta$ values, values greater than 1 are preferred. For instance, for processing 2K resolution images with a patch size of $128 \times 128$, $\epsilon = \zeta$ proves effective. Establishing an empirical relation between $\zeta$ and $\epsilon$ is a subject for future research. We have also observed that using the models trained with GD as the initial models in PatchGD can improve overall performance. However, there are instances where model training on GD is not possible. In such scenarios, one could use low-resolution models trained on GD or even conventional pre-trained models. Nevertheless, the effect of each of these choices needs to be thoroughly studied.

*On other tasks.* PatchGD can also be adapted for other tasks such as segmentation and generative modeling, among others. Our early results related to generative modeling look promising and pave way for future research. More details are presented in the supplementary material.

## 5 Conclusions

In this paper, we introduced Patch Gradient Descent (PatchGD), a novel training strategy for deep learning models that effectively handles large images even with limited GPU memory. PatchGD updates the model using partial image fractions, ensuring comprehensive context coverage over multiple steps. Through various experiments, we demonstrated the superior performance of PatchGD compared to standard gradient descent, both in handling large images and operating under low memory conditions. The presented method and

experimental evidence highlight the significance of PatchGD in enabling existing deep learning models to effectively process large images without compute memory limitations.

## 6    Limitations

While our numerical experiments have showcased the effectiveness of PatchGD, there are still limitations in terms of comprehensively understanding its generalization and stability. Additionally, our method's relative slowness compared to standard gradient descent is a minor drawback, particularly when real-time training is crucial. However, this limitation does not affect the inference speed, making it a bottleneck only in specific scenarios prioritizing real-time training.

*Gradient bias in PatchGD.* PatchGD introduces gradient bias in both the forward and backward passes, unlike methods such as activation checkpointing and activation offloading. During the forward pass, the bias arises because the classifier operates on stale z-vectors, which are derived from previous iterations. This results in suboptimal feature representations since the z-vectors do not accurately reflect the latest model updates. Unlike activation checkpointing or offloading, which recompute or store exact intermediate activations, PatchGD's dependence on these delayed z-vectors can lead to discrepancies between the computed and true activations.

In the backward pass, gradient bias occurs because some z-vectors do not propagate gradients. This incomplete gradient flow results from PatchGD's strategy of updating only a subset of z-vectors during each iteration. Additionally, due to overlapping receptive fields, neighboring patches can influence these z-vectors, leading to an uneven gradient propagation and an approximation that deviates from the true gradient.

To mitigate these biases, several strategies can be employed. Using smaller patch sizes reduces the forward pass bias by ensuring that z-vectors are updated more frequently, thereby decreasing the staleness effect. Introducing overlapping patches helps in capturing more accurate gradients by minimizing boundary effects and ensuring more uniform gradient propagation. Incorporating momentum in stochastic gradient descent (SGD) can help average out the bias over multiple iterations by leveraging historical gradient information to smooth out the noise introduced by the gradient bias.

Empirical evaluations show that while PatchGD offers significant memory savings, the introduced gradient bias results in noisier gradient updates. However, this bias does not significantly impact overall training performance and convergence. The benefits of reduced memory usage and the ability to train larger models with PatchGD outweigh the impact of gradient bias. We acknowledge the presence of this bias and recommend further studies to quantify and refine these strategies, enhancing the effectiveness of PatchGD in training large-scale models efficiently.

## 7    Future work

This paper has established the foundational concept of patch gradient descent to enable training CNNs using very large images and even when only limited GPU memory is available for training. The results as well as insights presented in the paper open doors to several novel secondary research directions that could be interesting in terms of improving the efficacy as well as the acceptance of the presented method in a broader scientific community. We list some such directions here.

- *Scaling to gigapixel images at small compute memory.* An ambitious but very interesting application of PatchGD would be to be able to process gigapixel images with small GPU memory. We can clearly envision this with PatchGD but with additional work. One important development needed is to extend the PatchGD learning concept to multiple hierarchical **Z** blocks, thereby sampling patches from the outer block to iteratively fill the information in the immediate inner $Z$ block and so on.

- *Enhanced receptive field.* So far, PatchGD has been looked at only in the context of being able to handle very large images. However, a different side of its use is that with almost the same architecture, it builds a smaller receptive build, thereby zooming in better. We speculate that in this context,

PatchGD could also help in building better discriminative models with lighter CNN architectures. Clearly, this would be of interest to the deep learning community and needs to be explored.

## Broader Impact

The broader impact of this work lies particularly in its potential to extend the capability of deep learning models. By addressing the challenge of training models on large-scale images with limited computational resources, our approach opens up opportunities for researchers and practitioners with constrained hardware setups to tackle complex problems in healthcare, agriculture, and environmental monitoring, where high-resolution images play a crucial role in decision-making processes. Moreover, our approach can contribute to reducing the environmental footprint of deep learning by enabling efficient training on low-power devices, thus promoting sustainability in the development and deployment of deep learning models. In summary, our work has the potential to empower diverse communities, drive sustainable development, and accelerate scientific progress. It is essential to approach these advancements with a conscientious mindset, taking into account the broader societal impact and proactively working towards an inclusive and responsible deployment of deep learning technologies. With our work, it is also important to address the potential risks and challenges. Issues related to data privacy, bias, and fairness should be carefully addressed to prevent any unintended negative consequences. Additionally, the potential for misuse or malicious applications of deep learning models should be acknowledged and proactively addressed through robust security measures and ethical guidelines.

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

## Supplementary Material

## A  Datasets

### A.1  PANDA

The Prostate cANcer graDe Assessment Challenge Bulten et al. (2022) consists of one of the largest publically available datasets for Histopathological images which scale to a very high resolution. It is important to mention that we do not make use of any masks as in other aforementioned approaches. Therefore, the complete task boils down to taking an input high-resolution image and then classifying them into 6 categories based on the International Society of Urological Pathology (ISUP) grade groups. There are a total of 10.6K images which are split into train and test sets in the ratio 80:20.

## A.2 UltraMNIST

This is a synthetic dataset generated by making use of the MNIST digits. For constructing an image, 3-5 digits are sampled such that the total sum of digits is less than 10. Thus an image can be assigned a label corresponding to the sum of the digits contained in the image. Each of the 10 classes from 0-9 has 1000 samples making the dataset sufficiently large. Note that the variation used in this dataset is an adapted version of the original data presented in Gupta et al. (2022), with background noise removed so that any shortcut learning is avoided Geirhos et al. (2020). Since the digits vary significantly in size and are placed far from each other, this dataset fits well in terms of learning semantic coherence in a image. Moreover, it poses the challenge that downscaling the images leads to a significant loss of information. While even higher resolution could be chosen, we later demonstrate that the chosen image size is sufficient to demonstrate the superiority of PatchGD over the conventional gradient descent method.

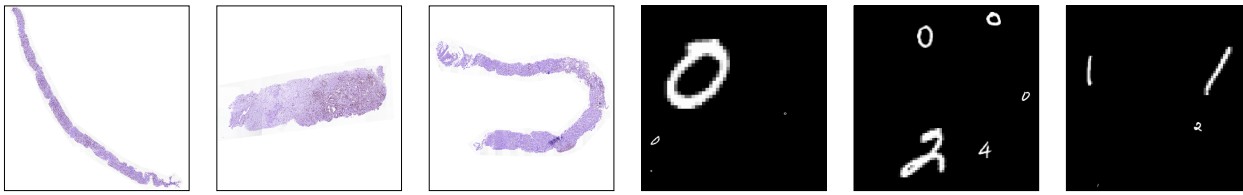

Figure 4: Sample PANDA and UltraMNIST dataset images used for training PatchGD.

## A.3 TCGA

The TCGA-NSCLC dataset, known as The Cancer Genome Atlas-Non-Small Cell Lung Cancer, encompasses two distinct types of lung cancer: Lung Adenocarcinoma (LUAD), with 522 cases, and Lung Squamous Cell Carcinoma (LUSC), with 504 cases, with a total number of image files 3220. The data was split in a stratified manner using the patient cases, into train and test set in the ratio 80:20, making sure there is no data leakage from train to test. The whole slide images are used to evaluate the performance of baseline and PatchGD in classifying the lung cancer subtypes.

# B Training Methodology and Hyperparameters

For Tables 1,2,3,5,7,8,9,10,11 presented in the main paper, all models are trained for 100 epochs with Adam optimizer and a peak learning rate of 1e-3. A learning rate warm-up for 2 epochs starting from 0 and linear decay for 98 epochs till half the peak learning rate was employed. The latent classification head consists of 4 convolutional layers with 256 channels in each. We perform gradient accumulation over inner iterations for better convergence, in the case of PANDA. To verify if results are better, not because of an increase in parameters (coming from the classification head), baselines are also extended with a similar head. GD*, for MobileNetV2 on UltraMNIST, refers to the baseline extended with this head.

In the case of low memory, as demonstrated in the UltraMNIST experiments, the original backbone architecture is trained separately for 100 epochs. This provides a better initialization for the backbone and is further used in PatchGD as mentioned in Tables 1 and 2.

For baseline in PANDA at 2048 resolution, another study involved gradient accumulation over images, which was done for the same number of images that can be fed when the percent sampling is 10% i.e. 14 times since a 2048x2048 image with a patch size of 128 and percentage sampling of 10 percent can have a maximum batch size of 14 under 16GB memory constraint. That is to say, the baseline can virtually process a batch of 14 images. This, however, was not optimal and the peak accuracy reported was in the initial epochs due to the loading of the pre-trained model on the lower resolution after which the metrics remained stagnant (accuracy: 32.11%, QWK:0.357).

For Table 4 presented in the main paper, we use the training strategies as mentioned in the respective works. The training strategy on TCGA is similar to what is employed on the PANDA dataset in Table 5.

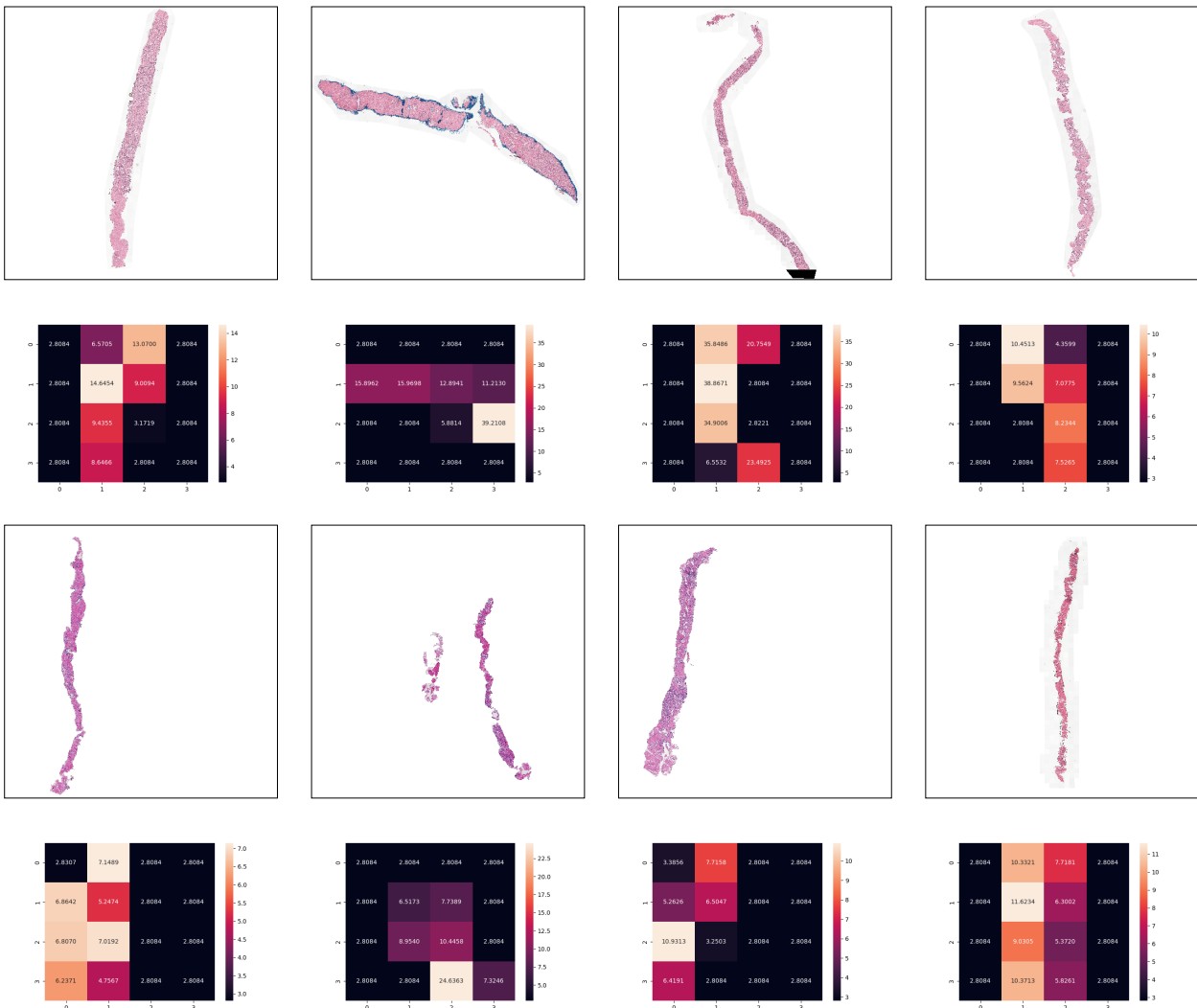

Figure 5: Sample PANDA images along with their latent space $Z$. It can be seen that the latent space clearly acts as a rich feature extractor.

For the ImageNet experiments in Table 6, we follow the exact training recipe as given in Touvron et al. (2021). This includes a 300 epoch training regime with cosine decay and a combination of multiple image augmentations[4].

**Convergence of baseline models.** For all the baseline experiments reported in the paper, we have also investigated extended training of the baseline to match the training time of the corresponding PatchGD experiments. However, it has been consistently observed that the configurations reported in the paper are the most optimal and the baseline converged within the initial 100 epochs for all the configurations. This clearly confirms that the gain reported by PatchGD is not due to the additional training time associated with this method.

## C  On other tasks

*Generative modeling.*  PatchGD can be used for generating large-scale images with a broad semantic context, which can be beneficial for data augmentation in fields such as deep learning for medical imaging. Early

---

[4]See Table 9. of Touvron et al. (2021)

Table 12: Comparison with normalization techniques at 2048 image size and 48GB memory constraint with Resnet50 backbone.

| Method | Batch Size | Setting | Accuracy % |
|---|---|---|---|
| BatchNorm | 6 | - | 49.4 |
| GroupNorm | 6 | Groups = 32 | 50.3 |
| Grad. Acc. | 5 | Steps = 11 | 44.1 |
| PatchGD | 56 | | 56.2 |

Table 13: Influence of different number of gradient accumulation steps $\epsilon$ on the performance of PatchGD.

| Model | Dataset | Memory | Image size | Patch size | $\epsilon$ | Accuracy |
|---|---|---|---|---|---|---|
| MobileNetv2 | UltraMNIST | 16 GB | 512 | 256 | 1 | **83.7** |
| MobileNetv2 | UltraMNIST | 16 GB | 512 | 256 | 2 | 81.5 |
| MobileNetv2 | UltraMNIST | 16 GB | 512 | 256 | 4 | 81.1 |
| Resnet50 | PANDA | 4GB | 512 | 64 | 1 | 41.9 |
| Resnet50 | PANDA | 4GB | 512 | 64 | 8 | **50.5** |
| Resnet50 | PANDA | 4GB | 512 | 64 | 32 | 45.0 |
| Resnet50 | PANDA | 48GB | 4096 | 256 | 8 | 56.9 |
| Resnet50 | PANDA | 48GB | 4096 | 256 | 32 | **59.7** |

results using StyleGAN-2 on the CIFAR-10 dataset showed that our method generated patches of $16 \times 16$ which were stitched together and analyzed by the discriminator, leading to a comparable FID score of 6.3 to the standard GD's FID score of 6.1. We believe this small performance gap can be eliminated with hyperparameter optimization. We consider that the potential of PatchGD in generative modeling can be maximized by generating large images with various semantic contexts, although this needs to be explored further.

*PatchGD for segmentation.* We discuss here how PatchGD can be used for tasks such as segmentation or any other encoder-decoder tasks We have discussed generative modeling already, and since the setup would be something similar, we present here an understanding of how the PatchGD formulation would unfold for tasks such as segmentation. For the task of segmentation as well, we have two sets of weights $\theta_1$ and $\theta_2$ that constitute the encoder and the decoder, respectively. Here, the encoder generates a $Z$-block and the decoder is used to generate the segmentation map from the $Z$-block. Similar to the classification problem, PatchGD operates on each image over a course of multiple inner iterations. At each inner iteration, patches are sampled from image $x$ and accordingly passed through and the output is then used to update the respective parts of $Z$. Further, $k$ $c$-dimensional vectors are sampled from $Z$ and passed through the decoder to generate mask patches that are used to update parts of the segmentation map $y$, and the process is repeated. Note that similar to $Z$–filling, this process also requires $y$-filling before the model updates of the encoder and decoder are performed over patches. For this purpose, we can first train a segmentation model on lower-resolution images of the chosen task and then use its encoder and decoder, and starting models for the PatchGD learning process.

## D   Comparison with normalization techniques

Batch normalization methods also influence the covergence of deep learning models at low batch sizes. However, PatchGD outperforms these techniques as well and we present a comparison is presented in Table 12.

## E   Gradient Accumulation Study

We also highlight an ablation study on the effect of changing the gradient accumulation steps $\epsilon$ as presented in Table 13. The gradients are accumulated and weights are updated only after $\epsilon$ steps. The ablations were

conducted for different epsilon settings, image and patch sizes, and memory constraints. We found that for smaller patch sizes, employing gradient accumulation steps greater than 1 is essential, with significant gains observed as the patch size to image size ratio decreases. Despite this promising trend, $\epsilon$ remains a hyperparameter requiring further tuning. Moreover, exploring the nuanced relationship between accuracy and steps is an essential aspect for future investigation in optimizing PatchGD. In case of UltraMNIST dataset at 512 image size, best performance is observed at $\epsilon = 1$ for a patch size of 256. For PANDA two variations were tried for image size 512 and image size 4096 with best results obtained at 8 and 32 respectively.

## F    Applications in Time Series Classification

Extending the concept of PatchGD to the 1-dimensional case, we find the application in time series classification. For this task, we take the example of UCI Human Activity Recognition Dataset. A set of 9 inertial signals at 128 unique time stamps are used to predict the action being executed (sitting, walking, etc.). For the baseline model, we use a basic 1-d Convolutional Network with 64 kernels each of size 3 and a linear layer at the end which achieves an accuracy of 88.9%. The model is trained using Adam as an optimizer with a constant learning rate of 1e-3 for 30 epochs with 32 batch size. The counterpart PatchGD-inspired approach involved the same 1-d convolutional network as the encoder with an intermediate latent vector, with other common hyperparameters being kept the same. The time series is broken into chunks temporally, each chunk being of length 16. Each inner iteration consists of sampling 25% of the total chunks with gradient updates enabled. The model is updated at the final iteration. Impressively, the approach achieves similar accuracy of 88.5%. The results are promising and yet again demonstrate the wide application to other tasks where PatchGD can be applied. Although this needs to be investigated further.

