# OpenReview forum: "Pushing the Limits of Gradient Descent for Efficient Learning on Large Images"
_TMLR — Accepted by TMLR_

### Review · Reviewer_ngg3 · 2024-03-18

**Summary Of Contributions:**

This work proposes patch gradient descent, a simple method of dividing gradient descent updates of large data into patches by using positional embeddings to locate each patch in the overall input.
The authors demonstrate the superiority of their patch-based method over standard gradient descent using the original image sizes, including when using gradient accumulation.
The authors also provide results for multiple model architectures and datasets, providing more extensive proof for their conclusions for multiple GPU memory budgets.

**Audience:**

Yes

**Claims And Evidence:**

No

**Requested Changes:**

The experiments require comparisons with the current SOTA methods for fair comparison. Although the authors have demonstrated superiority over naïvely using large images during training, it is difficult to conclude that they have demonstrated superiority over other methods.

**Strengths And Weaknesses:**

The authors provide a simple method of learning for very large inputs by splitting the input data along patches during training. They also show conclusively that their method outperforms naïve large-input training, including when gradient accumulation is applied.

The major weakness of the paper is that the only comparison is with naïve training as opposed to methods more specialized for large image training. Even in the case of the PANDA challenge, the QWK score of 0.730 presented by the authors is too far below the 0.862 score achieved in the original challenge to be considered a fair comparison. More rigorous comparisons with SOTA baselines are necessary.

---

> ### Author Response · Authors · 2024-06-21
> **Response to Reviewer ngg3**
>
> **W1: The only comparison is with naïve training as opposed to methods more specialized for large image training. In the case of the PANDA challenge, the QWK score of 0.730 presented by the authors is too far below the 0.862 score achieved in the original challenge**
>
> We understand the reviewer's concern regarding the reported performance of PatchGD in our experiment. It is true that our reported performance is lower than what has been observed in the challenge. However, it is important to note that our experiments were conducted under memory-constrained settings and were not intended to compete on the challenge leaderboard.
>
> Typically, solutions presented in such challenges involve several layers of model complexity, multi-stage networks, pre- and post-processing of results, and ensembles of solutions. In this paper, our aim is not to develop a state-of-the-art solution for the PANDA classification task. Instead, our objective is to use the PANDA classification task to demonstrate that larger images can be trained on constrained GPU settings. This is why we compare our solution with conventional training approaches.
>
> Additionally, we could not identify the specific solution referred to with the score of 0.862, as the leaderboard's top scores exceed 0.93. Our focus remains on showcasing the efficiency of PatchGD in resource-limited scenarios.
>
> Nevertheless, based on the reviewer’s recommendation, we present multiple comparisons of PatchGD on SOTA benchmarks.
>
> **Comparison with SOTA benchmark and method**: To demonstrate the efficacy of PatchGD on established large image benchmarks, we study the task of TCGA-NSCLC dataset classification and compare our solution with popular approaches such as HIPT and CLAM-SB, among others. Related results are presented in the table below. Note here that the HIPT model uses a 3-stage Transformer model with a ViT backbone which is pre-trained on an external large-scale histopathological dataset first and then fine-tuned on TCGA-NSCLC data. Further, it uses the images at a gigapixel scale. Further, CLAM-SB uses a multistage processing approach where a segmentation map is first obtained, followed by creating embedding of small patches. An attention pooling is then used to assign weight to each patch which together are then served to a classification model. The other baselines listed in the Table below similarly also use multistage processing with additional pretraining done on external datasets for boosted discriminative power.
>
> | Method                       | Model                                      | Image Size | AUC | Standard Deviation |
> |------------------------------|--------------------------------------------|------------|--------------|------------------------|
> | Baseline-1                   | ConvNeXt-V2 Tiny                           | 224        | 78.0         | 3.7                    |
> | GCN-MIL (Zhao et al., 2020)  | VAE-GAN + Graph CNN                        |        | 83.1         | 3.4                    |
> | MIL (Lu et al., 2021)        | -                           |        | 89.2         | 4.2                    |
> | Baseline-2                   | ConvNeXt-V2 Tiny                           | 4096       | 90.4         | 4.3                    |
> | DS-MIL (Li et al., 2021)     | Patching + Resnet18 + Aggregation          |       | 92.0         | 2.4                    |
> | CLAM-SB (Lu et al., 2021)    | Patching + Resnet50 + Attention Clustering |       | 92.8         | 2.1                    |
> | HIPT (Chen et al., 2022)     | Patching + 3 × Transformer                 |       | 95.2         | 2.1                    |
> | PatchGD                      | ConvNeXt-V2 Tiny                           | 4096       | 97.0         | 1.7                    |
>
> For our study on PatchGD, we employed a lightweight ConvNeXt-Tiny model without any pretraining on external histopathological datasets. Despite using only 4K images, no external pretraining, and a simple tiny backbone, PatchGD outperformed HIPT and other methods designed for handling large images, demonstrating its superiority. We believe that incorporating pre-training on external data, as done in HIPT, could further enhance the performance of the PatchGD solution. This discussion and the results are added on pages 10-11 of the updated draft (results in Table 5).

---

> > ### Author Response · Authors · 2024-06-21
> > **Response to Reviewer ngg3**
> >
> > **PatchGD on ImageNet.** To understand how PatchGD works with natural images, we study its performance on ImageNet dataset for different choices of number of classes as well as number of samples per class. We conduct these experiments using DeiT-Tiny transformer architecture and the results are reported in the table below. To study the effect of the number of samples, we fix classes to 25. Interestingly, we observe that PatchGD outperforms the standard training approach by around 2% accuracy, a significant improvement in the context of ImageNet training. We further examined the performance of PatchGD across different numbers of classes, keeping the number of samples per class fixed at 100. Interestingly, PatchGD outperformed the baseline approach when dealing with fewer classes. However, when the number of classes increased to 500, the baseline method performed better. This discrepancy arises because, for low-resolution images such as those in the ImageNet dataset, the small information loss at the edges of the patches becomes significant when there are many classes and limited samples per class. Our initial findings suggest that this issue can be mitigated to some extent by using overlapping patches, although this increases computational demands. Nonetheless, our observations indicate that PatchGD is the preferred choice for natural images in low-data regimes.
> >
> > | # Classes | # Samples / Class | Baseline Accuracy (%) 224 | Baseline Accuracy (%) 384 | Baseline Accuracy (%) 512 | PatchGD Accuracy (%) 512 |
> > |-----------|-------------------|---------------------------|---------------------------|---------------------------|--------------------------|
> > | 25        | 100               | 85.76                     | 88.68                     | 88.72                     | 90.74                    |
> > | 25        | 200               | 88.41                     | 90.32                     | 90.16                     | 92.12                    |
> > | 25        | 500               | 90.08                     | 92.00                     | 92.18                     | 93.14                    |
> > | 25        | 1000              | 91.28                     | 93.12                     | 93.20                     | 95.44                    |
> > | 10        | 100               | -                         | 85.40                     | -                         | 88.40                    |
> > | 25        | 100               | -                         | 88.68                     | -                         | 90.74                    |
> > | 100       | 100               | -                         | 76.90                     | -                         | 78.20                    |
> > | 500       | 100               | -                         | 73.65                     | -                         | 70.82                    |

---

> ### Author Response · Authors · 2024-06-21
> **Response to Reviewer ngg3**
>
> **Comparison with Activation Checkpointing and Offloading:** Alternate methods to train bigger models on smaller GPUs include activation checkpointing and offloading. We have now added a comparison with these approaches and the results are shown in the table below. We present the comparison of a ResNet50 architecture on the PANDA dataset at two different image resolutions on an NVIDIA 16 GB L4 graphics card. For gradient checkpointing, we employ chunk sizes of 4 and 6. PatchGD outperforms checkpointing and offloading approaches, particularly with 2K resolution images, where the margin of superiority is significantly larger. Under the selected memory constraint at this resolution, both baseline methods can only handle a maximum batch size of 4 per iteration, with activation offloading managing only 2. In contrast, PatchGD can handle batch sizes of 14. For smaller images, all methods can increase the batch size, but PatchGD still delivers the best performance. This demonstrates that PatchGD is more effective in utilizing GPU resources. Additionally, it is worth noting that PatchGD, checkpointing, and offloading are orthogonal methods and can be combined to fit even larger models on smaller GPU resources. This discussion and the table are added to page 12 of the updated draft (please see Table 8).
>
> | Method                             | Image Size | Batch Size | Peak Memory (GB) | Training time / image / iteration (ms) | Accuracy |
> |------------------------------------|------------|------------|------------------|----------------------------------------|----------|
> | Baseline                           | 2048       | 1          | 8.6              | 716                                    | 34.8     |
> | Activation Off-loading             | 2048       | 2          | 14.6             | 2603                                   | 42.1     |
> | Gradient Checkpointing, chunks=4   | 2048       | 3          | 14.1             | 1007                                   | 46.0     |
> | Gradient Checkpointing, chunks=6   | 2048       | 4          | 13.1             | 998                                    | 48.0     |
> | PatchGD                            | 2048       | 14         | 15.1             | 930                                    | 56.2     |
> | Baseline                           | 512        | 27         | 14.6             | 42                                     | 44.4     |
> | Activation Off-loading             | 512        | 32         | 14.9             | 161                                    | 46.8        |
> | Gradient Checkpointing, chunks=4   | 512        | 52         | 14.8             | 59                                     | 46.2        |
> | Gradient Checkpointing, chunks=6   | 512        | 72         | 14.6             | 59                                     | 44.7        |
> | PatchGD                            | 512        | 200        | 14.7             | 79                                     | 52.1     |

---

### Review · Reviewer_QP2J · 2024-03-30

**Summary Of Contributions:**

The paper presents PatchGD, a memory-efficient method for training CNNs on large images. The basic idea is to split the network into a patch-wise feature extractor and a final classifier, and to keep a buffer Z that stores the outputs of the feature extractor. For one batch of patches, the feature extractor is run and the corresponding vectors in Z are updated, before the classifier is run on the full Z (which also contains some stale feature vectors). The backprop is done, propagating gradients only through the z-vectors most recently computed.

**Audience:**

Yes

**Broader Impact Concerns:**

no concerns

**Claims And Evidence:**

Yes

**Requested Changes:**

The paper should really include a discussion on methods like checkpointing, FSDP, and activation offloading. It should also discuss and perhaps study gradient bias, which PatchGD has but these methods don't. Finally, it should benchmark compute and memory performance as well as accuracy vs checkpointing, and activation offloading.

**Strengths And Weaknesses:**

The paper shows convincingly that the method reduces memory and gives good results. My main concern is that the paper does not contain adequate baselines for memory-efficient training methods. For example I could not find any mention of checkpointing, which is a standard trick to reduce memory consumption. Similarly, the authors don't mention FullyShardedDataParallel (FSDP), pipeline parallelism, or simply offloading activations to CPU memory during the forward pass. In my view the authors should at least benchmark against checkpointing and activation offloading.

An important reference that is missing: "ZeRO: Memory Optimizations Toward Training Trillion Parameter Models"

Unlike the mentioned methods, PatchGD introduces bias to the gradients. This comes both from the forward pass (because the classifier is run on stale z-vectors) and the backward pass (because some z-vectors don't propagate a gradient, while pixels in neighbouring patches still could influence those z-vectors due to overlapping receptive fields). The paper should discuss gradient bias.

Some details are missing. E.g. it is not clear to me what the memory cost of the Z-buffer is, though it is probably small. I could not find information on where these buffers are stored. Are they in GPU memory?

---

> ### Author Response · Authors · 2024-06-21
> **Response to Reviewer QP2J**
>
> **W1, R1: Paper does not contain adequate baselines for memory-efficient training methods. The paper should really include a discussion on methods like checkpointing, FSDP, and activation offloading.**
>
> We have now extended the Related work section to include a discussion related to model parallelism, activation checkpointing as well as activation offloading. Details can be found on Page 4 of the updated draft.
>
> We have also added a comparison with Activation Checkpointing and Activation Offloading and the results are shown in the table below.  We present the comparison of a ResNet50 architecture on the PANDA dataset at two different image resolutions on an NVIDIA 16 GB T4 graphics card. For gradient checkpointing, we employ chunk sizes of 4 and 6. PatchGD outperforms checkpointing and offloading approaches, particularly with 2K resolution images, where the margin of superiority is significantly larger. Under the selected memory constraint at this resolution, both baseline methods can only handle a maximum batch size of 4 per iteration, with activation offloading managing only 2. In contrast, PatchGD can handle batch sizes of 14. For smaller images, all methods can increase the batch size, but PatchGD still delivers the best performance. This demonstrates that PatchGD is more effective in utilizing GPU resources. Additionally, it is worth noting that PatchGD, checkpointing, and offloading are orthogonal methods and can be combined to fit even larger models on smaller GPU resources. This discussion and the table are added to page 12 of the updated draft (please see Table 8).
>
> | Method                             | Image Size | Batch Size | Peak Memory (GB) | Training time / image / iteration (ms) | Accuracy |
> |------------------------------------|------------|------------|------------------|----------------------------------------|----------|
> | Baseline                           | 2048| 1| 8.6| 716| 34.8|
> | Activation Off-loading             | 2048| 2| 14.6| 2603| 42.1|
> | Gradient Checkpointing, chunks=4   | 2048| 3| 14.1| 1007| 46.0|
> | Gradient Checkpointing, chunks=6   | 2048| 4| 13.1| 998| 48.0     |
> | PatchGD                            | 2048       | 14         | 15.1             | 930                                    | 56.2     |
> | Baseline                           | 512        | 27         | 14.6             | 42| 44.4     |
> | Activation Off-loading             | 512        | 32         | 14.9             | 161| 46.8        |
> | Gradient Checkpointing, chunks=4   | 512| 52| 14.8| 59| 46.2        |
> | Gradient Checkpointing, chunks=6   | 512| 72| 14.6| 59| 44.7        |
> | PatchGD                            | 512| 200| 14.7| 79| 52.1     |
>
> **W2: An important reference that is missing: "ZeRO: Memory Optimizations Toward Training Trillion Parameter Models"**
>
> Thank you for pointing us to this important paper. We have now included it in the related work part of the updated draft.

---

> > ### Author Response · Authors · 2024-06-21
> > **Response to Reviewer QP2J**
> >
> > **W3: It should also discuss and perhaps study gradient bias, which PatchGD has but these methods don't.**
> >
> > Thank you for the remark. We agree that the issue of gradient bias in PatchGD should be discussed in the paper, and we have now added it in Section 6 of the paper under limitations tag.
> >
> > PatchGD introduces gradient bias in both the forward and backward passes, unlike methods such as activation checkpointing and activation offloading. During the forward pass, the bias arises because the classifier operates on stale z-vectors, which are derived from previous iterations. This results in suboptimal feature representations since the z-vectors do not accurately reflect the latest model updates. Unlike activation checkpointing or offloading, which recompute or store exact intermediate activations, PatchGD’s dependence on these delayed z-vectors can lead to discrepancies between the computed and true activations.
> >
> > In the backward pass, gradient bias occurs because some z-vectors do not propagate gradients. This incomplete gradient flow results from PatchGD's strategy of updating only a subset of z-vectors during each iteration. Additionally, due to overlapping receptive fields, neighbouring patches can influence these z-vectors, leading to an uneven gradient propagation and an approximation that deviates from the true gradient.
> >
> > To mitigate these biases, several strategies can be employed. Using smaller patch sizes reduces the forward pass bias by ensuring that z-vectors are updated more frequently, thereby decreasing the staleness effect. Introducing overlapping patches helps in capturing more accurate gradients by minimizing boundary effects and ensuring more uniform gradient propagation. Incorporating momentum in stochastic gradient descent (SGD) can help average out the bias over multiple iterations by leveraging historical gradient information to smooth out the noise introduced by the gradient bias.
> >
> > Empirical evaluations show that while PatchGD offers significant memory savings, the introduced gradient bias results in noisier gradient updates. However, this bias does not significantly impact overall training performance and convergence. The benefits of reduced memory usage and the ability to train larger models with PatchGD outweigh the impact of gradient bias. We acknowledge the presence of this bias and recommend further studies to quantify and refine these strategies, enhancing the effectiveness of PatchGD in training large-scale models efficiently.
> >
> > **W4: Some details are missing. E.g. it is not clear to me what the memory cost of the Z-buffer is, though it is probably small. I could not find information on where these buffers are stored. Are they in GPU memory?**
> >
> > Z-buffer is always part of the computational graph and resides in GPU memory. For an image size of 4096 with a 256 patch size and a 2048 embedding size, the Z-buffer is of shape 16 x 16 x 2048, taking up just ~2MB of space on the GPU. The Z-buffer is initialized using the latest model weights at the start of each batch and is maintained in GPU memory throughout the inner iterations of that batch. Once these iterations are completed, the Z-buffer is emptied before processing the next batch. This efficient management minimizes latency and ensures computational efficiency, making PatchGD an effective method for memory management while still providing significant memory savings.

---

> > > ### Comment · Reviewer_QP2J · 2024-07-07
> > > **Official comment**
> > >
> > > Thanks for the extensive updates. These sufficiently address my concerns.

---

### Review · Reviewer_Epb9 · 2024-06-11

**Summary Of Contributions:**

This work proposed PatchGD, a strategy for training convolutional networks for image recognition with very large image resolutions. The approach splits the ConvNet into two subnetworks, backbone and head. Specifically, the high resolution image is partitioned into a set of non-overlapping patches (similar to patchification in ViT); each patch is then independently projected by the backbone network into an embedding vector to construct a "Z-block," which is a latent 2D feature map. The convolutional head then operates on the entire Z-block to output an image class. To enable training on high resolution images while limiting activation memory, a large subset of embedding vectors in the Z-block are detached from the computational graph during backpropagation.

Experiments conducted with UltraMNIST and PANDA datasets using ResNet50, MobileNet-v2, ConvNeXt, and ViT models. The approach is found to work well with convolutional networks, but perhaps ill-suited in its current instantiation for transformer models. Numerical results demonstrate the ability to train on high resolution images by using gradient accumulation over several "inner loop" steps, where a subset of features in the Z-block are resampled in each inner iteration.

**Audience:**

Yes

**Broader Impact Concerns:**

Not applicable.

**Claims And Evidence:**

No

**Requested Changes:**

+ Please consider simplifying Algorithms 1 & 2, or at least increasing comments to improve clarity.
+ Could you clarify why the head $\theta_2$ is restricted to a conv-net.
+ Please consider simpler baselines for the ViT networks with GD (AdamW); e.g., a) detaching a subset of the input tokens from the computational graph, and/or b) randomly masking/dropping a subset of tokens, and/or c) increasing the patch size to reduce the number of tokens for high resolution images.
+ Please include GD baselines using gradient accumulation with the batch size found to work best for each model/dataset in memory-restricted settings, rather than reducing the batch size and degrading performance.
+ Please consider motivating PatchGD compared to standard tools such as activation checkpointing or model parallelism.
+ Please consider adding results on other datasets with well-established baselines for the considered architectures, e.g., ImageNet or CIFAR100.

**Strengths And Weaknesses:**

**Motivation**:
+ S1. The problem of training on higher resolution images is interesting and well motivated, especially in the context of high resolution medical images.

**Methodology**:
+ S2. The proposed method is also conceptually simple at its core, and seems like a natural approach for handling high resolution inputs.
- W1. However, the specific instantiation of the proposed approach in Algorithms 1 & 2 seems overly complex, with several several inner iterations per batch, each sampling a different subset of Z-block patches, and with several gradient accumulation steps composed with the inner loop.
- W2. The motivation for restriction head network to a convolutional architecture was not very clear to me.
- W3. While basic experiments combining PatchGD with a ViT network did not work well in Table 7, there are probably much simpler baselines to experiment with, especially since ViT networks already include a patchification; e.g., what about detaching a subset of the input tokens from the computational graph, or randomly masking/dropping a subset of tokens? what about increasing the patch size to reduce the number of tokens for high resolution images? how about hierarchically processing progressively larger regions?
- W4. The value of the proposed approach is not clear; it remains to be verified whether it is necessary to train at a high resolution for the entire duration, especially since recent results in CV suggest that you can reap most advantages of high resolution images by only increasing image resolution during the last few epochs of training (e.g., DINOv2). It is also not clear whether PatchGD is a necessary strategy when one can leverage tools such as activation checkpointing or model parallelism.

**Experiments**:
+ S3. Several convolutional architectures and image classification datasets were considered; including combining GD baselines with an additional convolutional head.
+ W5. Concerns about the nature of the results and the baselines. Firstly, adding an additional convolutional head to the GD baselines should not degrade performance as in Tables 1 & 2, suggesting perhaps that these experiments were conducted with suboptimal hyperparameters. Secondly, GD performance should not degrade with the GPU memory. You can maintain the same baseline using gradient accumulation, especially since PatchGD employs gradient accumulation.
+ W6. The experimental setting is somewhat contrived, and in this case, there is no reason not to include similar results on more standardized image classification datasets; e.g., ImageNet where you artificially constrain the GPU memory, and where well established baselines are available for the considered architectures.

**Clarity**:
- W7. The exposition, specifically the discussion of the method in Algorithms 1 and 2, and the associated text, could be improved to reduce reliance on cumbersome notation

---

> ### Author Response · Authors · 2024-06-22
> **Response to Reviewer Epb9**
>
> **W1, R1: However, the specific instantiation of the proposed approach in Algorithms 1 & 2 seems overly complex, Please consider simplifying Algorithms 1 & 2, or at least increasing comments to improve clarity.**
>
> We have now added additional comments in Algorithms 1 & 2 for improving the clarity of the algorithms.
>
> **W2, R2: The motivation for restriction head network to a convolutional architecture was not very clear to me.**
>
> The initial restriction of the head to a convolutional neural network (CNN) in our PatchGD setup was primarily for practical reasons. Our main goal was to get the entire PatchGD pipeline to work effectively. Starting with a CNN backbone and a CNN head provided the simplicity and consistency needed for initial development. This choice allowed us to focus on stabilizing the pipeline and ensuring its functionality without introducing additional complexities from more diverse architectures.
>
> We also conducted preliminary experiments with other architectures. For example, we tested attention heads, which showed slight performance improvements. These findings were reported in the additional study section of Section 4.2 under the heading 'On attention head module'. Building on these results, we have extended our study to include transformer backbones and examined how both CNN and transformer heads influence overall performance. The results of these experiments with transformer architectures are presented in Table 6, and the comparison between CNN and transformer heads is detailed in Table 7 of the updated draft.
>
> **W3, R3:  Consider simpler baselines for the ViT networks with GD**
>
> In our early experiments with transformer backbones, we initially used CNN heads, which we identified as a primary reason for the reduced model performance. In our updated work, we experimented with transformer backbones combined with transformer heads. This combination led to superior performance, even allowing PatchGD to surpass standard training on the ImageNet dataset. These results are discussed in detail in a later comment and are provided in Table 6 of the updated draft. Additionally, we present a comparison of how the transformer head performs better than a CNN head when using a transformer backbone in Table 7 of the updated draft.
>
> We appreciate the reviewer's suggestions for improving PatchGD with transformer networks. In our investigations, we have conducted preliminary studies in similar directions and briefly discuss the associated challenges here:
>
> **Detaching a Subset of Input Tokens or Randomly Masking/Dropping Tokens**: Both methods are inherently lossy and risk failing to capture discriminative features if critical tokens are dropped. Aligning this approach with PatchGD would imply dropping a significant portion, such as 75%, of the patches at every iteration, which can lead to unstable convergence.
>
> **Increasing Patch Size to Reduce the Number of Tokens**: While this approach reduces the number of tokens, it deteriorates the quality of individual tokens as more spatial information needs to be learnt in a smaller embedding space.
>
> **Hierarchically Processing Progressively Larger Regions**: This is a promising approach for future work. Constructing multiple Z-blocks at different depths in the network could potentially allow PatchGD to be applied to images of any size. This method could enable efficient processing by progressively integrating larger regions hierarchically.
>
> We thank the reviewer for their valuable feedback and suggestions, which help guide our future research directions in optimizing PatchGD with transformer architectures.
>
> **W4:  Recent results in CV suggest that you can reap most advantages of high resolution images by only increasing image resolution during the last few epochs of training (e.g., DINOv2)**
>
> To address this concern, we conducted experiments using ResNet50 on the PANDA dataset at two different resolutions. For 2K resolution training, we performed the initial training at 512 resolution before scaling up, and for 4K resolution training, we started at 2K resolution. The results are summarized in the table below.
>
> | Image Size | Scratch | Pre-trained on previous stage | PatchGD |
> | --- | --- | --- | --- |
> | 512 | 44.4 | - | 52.1 |
> | 2048 | 49.4 | 52.9 | 56.2 |
> | 4096 | 50.0 | 52.1 | 59.7 |
>
> The hierarchical approach of initially training at a lower resolution before increasing to a higher resolution did show improvements over the baseline. However, the performance gains achieved by PatchGD were substantially higher in both cases. These results demonstrate the significant advantages of using PatchGD for high-resolution image training. For other similar experiments, we consistently had similar observations. This validates the necessity and effectiveness of PatchGD in handling high-resolution image training, offering substantial gains over traditional hierarchical approaches.

---

> > ### Author Response · Authors · 2024-06-22
> > **Response to Reviewer Epb9**
> >
> > **R4: Please include GD baselines using gradient accumulation with the batch size found to work best for each model/dataset in memory-restricted settings**
> >
> > We understand and appreciate the reviewer's concern regarding the inclusion of gradient accumulation baselines. However, conducting a comprehensive hyperparameter sweep over gradient accumulation steps is extremely resource-intensive, as it requires full training runs for each setting. Moreover, any setup applied to the traditional approach can also be utilized for PatchGD, making it essential to ensure a fair comparison by applying the same gradient accumulation steps to the outer batches of PatchGD as well.
> >
> > Gradient accumulation is a lossy approach for batchnorm-based models and the better alternatives are Activaton Checkpointing and Offloading, for scaling up the batch size in a memory constrained setup. We conducted experiments for these approaches with ResNet50 on the PANDA dataset at 2K and 512 resolutions, incorporating different total batch sizes for the baseline and have been added in Table 8 of the updated draft. For the 2K resolution, the baseline score was 34.8, with the maximum performance achieved through extended batch size reaching 48.0, while PatchGD achieved a score of 56.1. For the 512 resolution, the baseline and PatchGD scores were 44.4 and 52.1, respectively, with the best gradient accumulation setup resulting in a score of 46.8. These experiments were conducted under a GPU memory constraint of 16 GB. It is important to note that PatchGD does not use gradient accumulation (or checkpointing or offloading) over the image batches. We speculate that the performance of PatchGD could be further improved with the addition of gradient accumulation.
> >
> > An interesting observation was that attempts to push gradient accumulation in the standard process to match the batch size of PatchGD resulted in significantly lower performance, even below the baseline. This is illustrated in Table 12 of the updated draft, where the resultant model's performance is substantially lower than the baseline. These results indicate that even with gradient accumulation, the standard training approach is inferior to PatchGD.
> >
> > **R5: Please consider motivating PatchGD compared to standard tools such as activation checkpointing or model parallelism.**
> >
> > We have now extended the Related work section to include a discussion related to model parallelism, activation checkpointing as well as activation offloading. Details can be found on Page 4 of the updated draft.
> >
> > **Comparison with Activation Checkpointing and Offloading:** Alternate methods to train bigger models on smaller GPUs include activation checkpointing and offloading. We have now added a comparison with these approaches and the results are shown in the table below. We present the comparison of a ResNet50 architecture on the PANDA dataset at two different image resolutions on an NVIDIA 16 GB T4 graphics card. For gradient checkpointing, we employ chunk sizes of 4 and 6. PatchGD outperforms checkpointing and offloading approaches, particularly with 2K resolution images, where the margin of superiority is significantly larger. Under the selected memory constraint at this resolution, both baseline methods can only handle a maximum batch size of 4 per iteration, with activation offloading managing only 2. In contrast, PatchGD can handle batch sizes of 14. For smaller images, all methods can increase the batch size, but PatchGD still delivers the best performance. This demonstrates that PatchGD is more effective in utilizing GPU resources. Additionally, it is worth noting that PatchGD, checkpointing, and offloading are orthogonal methods and can be combined to fit even larger models on smaller GPU resources. This discussion and the table are added to page 12 of the updated draft (please see Table 8).
> >
> > | Method                             | Image Size | Batch Size | Peak Memory (GB) | Training time / image / iteration (ms) | Accuracy |
> > |------------------------------------|------------|------------|------------------|----------------------------------------|----------|
> > | Baseline                           | 2048       | 1          | 8.6              | 716| 34.8     |
> > | Activation Off-loading             | 2048       | 2          | 14.6| 2603| 42.1     |
> > | Gradient Checkpointing, chunks=4   | 2048       | 3| 14.1| 1007| 46.0     |
> > | Gradient Checkpointing, chunks=6   | 2048       | 4| 13.1| 998| 48.0     |
> > | PatchGD                            | 2048       | 14         | 15.1| 930| 56.2     |
> > | Baseline                           | 512        | 27         | 14.6| 42| 44.4     |
> > | Activation Off-loading             | 512        | 32         | 14.9| 161| -        |
> > | Gradient Checkpointing, chunks=4   | 512        | 52| 14.8| 59| -        |
> > | Gradient Checkpointing, chunks=6   | 512        | 72| 14.6| 59| -        |
> > | PatchGD                            | 512        | 200        | 14.7| 79| 52.1     |

---

> ### Author Response · Authors · 2024-06-22
> **Response to Reviewer Epb9**
>
> **W6, R6: Please consider adding results on other datasets with well-established baselines for the considered architectures, e.g., ImageNet or CIFAR100.**
>
> Thank you for the remark. We understand the need for demonstrating the performance of PatchGD on standard benchmarks. In this regard, we have now considered two additional studies:
>
> **PatchGD on ImageNet.** To understand how PatchGD works with natural images, we study its performance on ImageNet dataset for different choices of number of classes as well as number of samples per class. We conduct these experiments using DeiT-Tiny transformer architecture and the results are reported in the table below. To study the effect of the number of samples, we fix classes to 25. Interestingly, we observe that PatchGD outperforms the standard training approach by around 2% accuracy, a significant improvement in the context of ImageNet training. We further examined the performance of PatchGD across different numbers of classes, keeping the number of samples per class fixed at 100. Interestingly, PatchGD outperformed the baseline approach when dealing with fewer classes. However, when the number of classes increased to 500, the baseline method performed better. This discrepancy arises because, for low-resolution images such as those in the ImageNet dataset, the small information loss at the edges of the patches becomes significant when there are many classes and limited samples per class. Our initial findings suggest that this issue can be mitigated to some extent by using overlapping patches, although this increases computational demands. Nonetheless, our observations indicate that PatchGD is the preferred choice for natural images in low-data regimes. These results are added in Table 6 of the updated draft.
>
> | # Classes | # Samples / Class | Baseline Accuracy (%) 224 | Baseline Accuracy (%) 384 | Baseline Accuracy (%) 512 | PatchGD Accuracy (%) 512 |
> |-----------|-------------------|---------------------------|---------------------------|---------------------------|--------------------------|
> | 25| 100| 85.76| 88.68| 88.72| 90.74|
> | 25| 200| 88.41| 90.32| 90.16| 92.12|
> | 25| 500| 90.08| 92.00| 92.18| 93.14|
> | 25| 1000| 91.28| 93.12| 93.20| 95.44|
> | 10| 100| -| 85.40| -| 88.40|
> | 25| 100| -| 88.68| -| 90.74|
> | 100| 100| -| 76.90| -| 78.20|
> | 500| 100| -| 73.65| -| 70.82|
>
> **Comparison with SOTA benchmark and method**: To demonstrate the efficacy of PatchGD on established large image benchmarks, we study the task of TCGA-NSCLC dataset classification and compare our solution with popular approaches such as HIPT and CLAM-SB, among others. Related results are presented in the table below. Note here that the HIPT model uses a 3-stage Transformer model with a ViT backbone which is pre-trained on an external large-scale histopathological dataset first and then fine-tuned on TCGA-NSCLC data. Further, it uses the images at a gigapixel scale. Further, CLAM-SB uses a multistage processing approach where a segmentation map is first obtained, followed by creating embedding of small patches. An attention pooling is then used to assign weight to each patch which together are then served to a classification model. The other baselines listed in the Table below similarly also use multistage processing with additional pretraining done on external datasets for boosted discriminative power.
>
> | Method                       | Model                                      | Image Size | Accuracy (%) | Standard Deviation (%) |
> |------------------------------|--------------------------------------------|------------|--------------|------------------------|
> | Baseline-1| ConvNeXt-V2 Tiny| 224        | 78.0         | 3.7                    |
> | GCN-MIL (Zhao et al., 2020)  | VAE-GAN + Graph CNN|        | 83.1         | 3.4                    |
> | MIL (Lu et al., 2021)| |        | 89.2         | 4.2                    |
> | Baseline-2| ConvNeXt-V2 Tiny| 4096       | 90.4         | 4.3                    |
> | DS-MIL (Li et al., 2021)     | Patching + Resnet18 + Aggregation|       | 92.0         | 2.4                    |
> | CLAM-SB (Lu et al., 2021)    | Patching + Resnet50 + Attention Clustering |       | 92.8         | 2.1                    |
> | HIPT (Chen et al., 2022)     | Patching + 3 × Transformer|       | 95.2         | 2.1                    |
> | PatchGD| ConvNeXt-V2 Tiny| 4096       | 97.0         | 1.7|
>
> For our study on PatchGD, we employed a lightweight ConvNeXt-Tiny model without any pretraining on external histopathological datasets. Despite using only 4K images, no external pretraining, and a simple tiny backbone, PatchGD outperformed HIPT and other methods designed for handling large images, demonstrating its superiority. We believe that incorporating pre-training on external data, as done in HIPT, could further enhance the performance of the PatchGD solution. This discussion and the results are added on pages 10-11 of the updated draft (results in Table 5).

---

> > ### Author Response · Authors · 2024-06-23
> > **Response to Reviewer Epb9**
> >
> > **W5: adding an additional convolutional head to the GD baselines should not degrade performance**
> >
> > When incorporating pre-trained weights, a model like ResNet50 is finely tuned to extract a diverse array of features from its training dataset, such as ImageNet. This tuning results in an optimal set of feature detectors that can generalize well across similar tasks. Adding additional convolutional layers to a pre-trained ResNet50 can disrupt this established feature hierarchy by introducing new parameters that modify the effective representation of learned features. Yosinski et al. (2014) demonstrated that transferring features even from the initial layers of a network can significantly impact performance, particularly when the new layers distort the original, beneficial feature mappings developed through pre-training.
> >
> > Yosinski, J., Clune, J., Bengio, Y., & Lipson, H. (2014). How transferable are features in deep neural networks? In Advances in neural information processing systems (pp. 3320-3328).

---

### Review · Reviewer_qfar · 2024-06-17

**Summary Of Contributions:**

The paper proposes a method to limit the memory requirements when training convolutional neural networks for image classification on large resolution images. The method assumes that the representation of different patches, extracted from the high resolution image, are roughly independent in the earlier layers of the network. Thus, at a given point in training, it avoids having to re-compute the representation of all patches of the image before a given layer, when the parameters haven't changed significantly (an hyperparameter decides how often they are recomputed). The paper tests the proposed method, PatchGD, and compares it against vanilla (S)GD on different academic datasets, with images of relatively high resolution (ranging from 512 to 4096 image size).

**Audience:**

Yes

**Broader Impact Concerns:**

No major ethical concerns.

**Claims And Evidence:**

No

**Requested Changes:**

- Provide clear total training and inference cost (either in a given hardware, or FLOPs) of the proposed method and the baseline, for a fair comparison.
- Include experiments with well stablished baselines, as mentioned in the weaknesses; and using well stablished benchmarks (e.g. ImageNet).

**Strengths And Weaknesses:**

**Strengths**
- The experiments show that the proposed method achieves higher accuracies than vanilla (S)GD, at different maximum memory limits, on two academic datasets of images of high resolution (UltraMNIST and PANDA).
- The paper includes additional (and limited, see next) experiments performed on the ImageNet100 dataset at a smaller resolution, showing that the method is applicable not only to high resolution images, but also on natural images of more standard image resolutions..

**Weaknesses**
- My main concern is that all comparison are done purely limiting the maximum amount of memory, but no details are provided regarding the training time. The proposed method has a couple of hyperparameters that can scale it cost significantly, compared to vanilla SGD. Namely, the number of inner iterations and the number of sampled patched in Algorithm 1. These hyperparameters have a significant impact in accuracy (see Table 8), but as I mentioned are completely neglected from a training time perspective. Suppose that to achieve the results reported in the paper, the total training time with PatchGD becomes m times larger than GD. What would the results of GD be if one trained simply for m times longer?
- Most of the experiments are done in rather small datasets (PANDA and UltraMNIST). The paper does include a few extra experiments on some other datasetts, but given that most of the experiments are conducted on the smaller ones, is not clear whether the proposed approach will generalise to more realistic scenarios.
- There are many baselines that are completely ignored from the paper. For instance, activation checkpointing / offloading, which is currently available and relatively easy to implement in most frameworks (e.g. PyTorch, JAX), or training in lower & mixed precision.
- Many state of the art works in image classification use Vision Transformers, instead of ConvNets. The paper includes some experiments using PatchGD with DeiT, but it's unclear how to use the proposed approach with a pure Transformer arquitecture.

---

> ### Author Response · Authors · 2024-06-20
> **Response to Reviewer qfar**
>
> **W1, R1: comparison are done purely limiting the maximum amount of memory, but no details are provided regarding the training time**
>
> Thank you for your remark. We want to emphasize that the baseline results in the paper are already optimized to their best possible configurations. Some experiment ensured that the baseline underwent extended training, matching the total training time with that of PatchGD. Despite this, the baselines consistently reached the best score within the first 100 epochs of training. A brief discussion on this has been added to Appendix B of the updated draft (highlighted in blue).
>
> Furthermore, PatchGD does not significantly increase training time. While it might seem that several inner iterations and fewer sampled patches would slow it down compared to the baseline method, PatchGD enables training with larger batch sizes, leading to fewer outer iterations. For example, as shown in the table below, with 2K images, the baseline's training time per iteration is 0.716 seconds, while PatchGD's is 0.93 seconds. The difference is not too big because the baseline processes a batch size of 1, whereas PatchGD processes a batch size of 14. As image size increases, this gap shrinks further.
>
> | Method                             | Image Size | Batch Size | Peak Memory (GB) | Training time / image / iteration (ms) | Accuracy |
> |------------------------------------|------------|------------|------------------|----------------------------------------|----------|
> | Baseline                           | 2048       | 1          | 8.6              | 716                                    | 34.8     |
> | Activation Off-loading             | 2048       | 2          | 14.6             | 2603                                   | 42.1     |
> | Gradient Checkpointing, chunks=4   | 2048       | 3          | 14.1             | 1007                                   | 46.0     |
> | Gradient Checkpointing, chunks=6   | 2048       | 4          | 13.1             | 998                                    | 48.0     |
> | PatchGD                            | 2048       | 14         | 15.1             | 930                                    | 56.2     |
> | Baseline                           | 512        | 27         | 14.6             | 42                                     | 44.4     |
> | Activation Off-loading             | 512        | 32         | 14.9             | 161                                    | 46.8        |
> | Gradient Checkpointing, chunks=4   | 512        | 52         | 14.8             | 59                                     | 46.2        |
> | Gradient Checkpointing, chunks=6   | 512        | 72         | 14.6             | 59                                     | 44.7        |
> | PatchGD                            | 512        | 200        | 14.7             | 79                                     | 52.1     |
>
> The difference in training time between the two approaches is higher for smaller images, as shown in the table. However, as mentioned earlier, we also experimented by letting the baseline train for longer durations, and it did not result in any further improvement. These results are added in Table 8 of the updated draft.
>
> Information related to throughput was already provided in the previous draft and is also described in Table 3 of the updated draft. In general, PatchGD throughput is less than the standard approach, however, this difference diminishes at large-size images. Nevertheless, the gain in accuracy is significantly higher compared to the added inference time incurred by PatchGD.
>
>
> **W3: There are many baselines that are completely ignored from the paper. For instance, activation checkpointing / offloading**
>
> *Comparison with Activation Checkpointing and Offloading:* We have now added a comparison with Activation Checkpointing and Activation Offloading and the results are shown in the table above.  We present the comparison for a ResNet50 architecture on the PANDA dataset at two different image resolutions on an NVIDIA 16 GB L4 graphics card. For gradient checkpointing, we employ chunk sizes of 4 and 6. PatchGD outperforms checkpointing and offloading approaches, particularly with 2K resolution images, where the margin of superiority is significantly larger. Under the selected memory constraint at this resolution, both baseline methods can only handle a maximum batch size of 4 per iteration, with activation offloading managing only 2. In contrast, PatchGD can handle batch sizes of 14. For smaller images, all methods can increase the batch size, but PatchGD still delivers the best performance. This demonstrates that PatchGD is more effective in utilizing GPU resources. Additionally, it is worth noting that PatchGD, checkpointing, and offloading are orthogonal methods and can be combined to fit even larger models on smaller GPU resources. This discussion and the table are added to page 12 of the updated draft (please see Table 8).

---

> ### Author Response · Authors · 2024-06-20
> **Response to Reviewer qfar**
>
> **W2: Most of the experiments are done in rather small datasets (PANDA and UltraMNIST).**
>
> *Comparison with SOTA benchmark and method:* To demonstrate the efficacy of PatchGD on established large image benchmarks, we study the task of TCGA-NSCLC dataset classification and compare our solution with popular approaches such as HIPT and CLAM-SB, among others. Related results are presented in the table below. Note here that the HIPT model uses a 3-stage Transformer model with a ViT backbone which is pre-trained on an external large-scale histopathological dataset first and then fine-tuned on TCGA-NSCLC data. Further, it uses the images at a gigapixel scale. Further, CLAM-SB uses a multistage processing approach where a segmentation map is first obtained, followed by creating embedding of small patches. An attention pooling is then used to assign weight to each patch which together are then served to a classification model. The other baselines listed in the Table below similarly also use multistage processing with additional pretraining done on external datasets for boosted discriminative power.
>
> | Method                       | Model                                      | Image Size | AUC | Standard Deviation|
> |------------------------------|--------------------------------------------|------------|--------------|------------------------|
> | Baseline-1                   | ConvNeXt-V2 Tiny                           | 224        | 78.0         | 3.7                    |
> | GCN-MIL (Zhao et al., 2020)  | VAE-GAN + Graph CNN                        |        | 83.1         | 3.4                    |
> | MIL (Lu et al., 2021)        | -                           |        | 89.2         | 4.2                    |
> | Baseline-2                   | ConvNeXt-V2 Tiny                           | 4096       | 90.4         | 4.3                    |
> | DS-MIL (Li et al., 2021)     | Patching + Resnet18 + Aggregation          |       | 92.0         | 2.4                    |
> | CLAM-SB (Lu et al., 2021)    | Patching + Resnet50 + Attention Clustering || 92.8         | 2.1                    |
> | HIPT (Chen et al., 2022)     | Patching + 3 × Transformer                 |       | 95.2         | 2.1                    |
> | PatchGD                      | ConvNeXt-V2 Tiny                           | 4096       | 97.0         | 1.7                    |

---

> > ### Author Response · Authors · 2024-06-20
> > **Response to Reviewer qfar**
> >
> > **R2: Include experiments with well stablished baselines, as mentioned in the weaknesses; and using well stablished benchmarks (e.g. ImageNet).**
> >
> > *PatchGD on ImageNet.* To understand how PatchGD works with natural images, we study its performance on ImageNet dataset for different choices of number of classes as well as number of samples per class. We conduct these experiments using DeiT-Tiny transformer architecture and the results are reported in the table below. To study the effect of the number of samples, we fix classes to 25. Interestingly, we observe that PatchGD outperforms the standard training approach by around 2% accuracy, a significant improvement in the context of ImageNet training. We further examined the performance of PatchGD across different numbers of classes, keeping the number of samples per class fixed at 100. Interestingly, PatchGD outperformed the baseline approach when dealing with fewer classes. However, when the number of classes increased to 500, the baseline method performed better. This discrepancy arises because, for low-resolution images such as those in the ImageNet dataset, the small information loss at the edges of the patches becomes significant when there are many classes and limited samples per class. Our initial findings suggest that this issue can be mitigated to some extent by using overlapping patches, although this increases computational demands. Nonetheless, our observations indicate that PatchGD is the preferred choice for natural images in low-data regimes.
> >
> > | # Classes | # Samples / Class | Baseline Accuracy (%) 224 | Baseline Accuracy (%) 384 | Baseline Accuracy (%) 512 | PatchGD Accuracy (%) 512 |
> > |-----------|-------------------|---------------------------|---------------------------|---------------------------|--------------------------|
> > | 25        | 100               | 85.76                     | 88.68                     | 88.72                     | 90.74                    |
> > | 25        | 200               | 88.41                     | 90.32                     | 90.16                     | 92.12                    |
> > | 25        | 500               | 90.08                     | 92.00                     | 92.18                     | 93.14                    |
> > | 25        | 1000              | 91.28                     | 93.12                     | 93.20                     | 95.44                    |
> > | 10        | 100               | -                         | 85.40                     | -                         | 88.40                    |
> > | 25        | 100               | -                         | 88.68                     | -                         | 90.74                    |
> > | 100       | 100               | -                         | 76.90                     | -                         | 78.20                    |
> > | 500       | 100               | -                         | 73.65                     | -                         | 70.82                    |

---

> > > ### Author Response · Authors · 2024-06-20
> > > **Response to Reviewer qfar**
> > >
> > > **W4:  it's unclear how to use the proposed approach with a pure Transformer architecture.**
> > >
> > > In our previous iteration of experiments, we had been experimenting with PatchGD where the backbone was a transformer architecture but the head was CNN and we could not achieve good convergence. However, we have observed that with the head comprising a small Transformer model, PatchGD couples well with Transformer backbones. Interestingly, we are not able to achieve superior performance and better convergence, even better than CNN backbones as expected. An example experiment is that on ImageNet where with PatchGD applied to Transformer, we achieve SOTA results. We have also performed a comparison of CNN and transformer heads for a transformer backbone and the results are presented in the table below (added as Table 7 in the updated draft).
> > >
> > > Details related to the training of the transformer model are now described in Appendix B. We follow the exact training recipe as given in Touvron et al (2021). This includes a 300 epoch training regime with cosine decay and a combination of multiple image augmentations as described in Table 9 of the paper of Touvron et al (2021). Further details related to the transformer head are added on page 12 of the updated draft (highlighted in blue). For the transformer head, we use a single multi-headed self-attention layer with with 2 heads each of 384 channels followed by a linear layer.
> > >
> > > | Method    | Resolution | Patch Size | Batch Size | Head        | Mem. (GB) | Accuracy % | QWK  |
> > > |-----------|------------|------------|------------|-------------|-----------|------------|------|
> > > | Baseline  | 512        | -          | 22         | -           | 16        | 48.4       | 0.596|
> > > | PatchGD   | 512        | 128        | 136        | CNN         | 16        | 44.9       | 0.576|
> > > | PatchGD   | 512        | 128        | 136        | Transformer | 16        | 48.7       | 0.599|
> > > | Baseline  | 2048       | -          | 4          | -           | 48        | 48.6       | 0.612|
> > > | PatchGD   | 2048       | 128        | 32         | CNN         | 48        | 48.9       | 0.589|
> > > | PatchGD   | 2048       | 128        | 32         | Transformer | 48        | 57.4       | 0.702|

---

### Decision · Action_Editor_Gb4w · 2024-08-13

**Recommendation:** Accept with minor revision

**Comment:**

This paper presents PatchGD, a method designed to reduce memory requirements when training CNNs on large-resolution images. The approach involves splitting the network into two parts, processing patches of the image independently, and using a buffer to store intermediate features. The buffer is used in its partial form in a stochastic manner for efficiency. The method is tested on various datasets and shows promising results in terms of memory efficiency and accuracy.

Reviewers highlighted several strengths, including the relevance of the problem and the novelty of the approach. However, they also raised concerns about the lack of comparison with other memory-efficient methods, the complexity of the proposed algorithm, and the limited scope of experiments. These concerns were adequately addressed in the rebuttal phase, where the authors provided additional baselines, clarified the methodology, and expanded the experiments.

Given the importance of memory efficiency in deep learning and the promising results shown by PatchGD, I believe this paper makes a valuable contribution to the field. While some limitations remain, such as the potential complexity of the approach and its generalizability to transformer architectures, these do not detract significantly from the overall contribution.

**Audience:**

The findings of this paper would be of interest to a significant portion of the TMLR audience, particularly those working on large-scale image classification tasks or developing methods for efficient deep learning. Memory efficiency is a critical concern in modern machine learning, especially with the growing use of high-resolution images and large models. This paper offers a novel approach that could inspire further research and development in this area.

**Claims And Evidence:**

Scaling machine learning models in terms of memory is an important problem and the paper presents a promising method for addressing this in the case of CNNs applied to large images. The claims made in the submission are supported by convincing evidence, particularly in terms of the method's ability to reduce memory usage during the training of convolutional neural networks (CNNs) on large images. The authors provide experimental results demonstrating that PatchGD can achieve comparable or better accuracy than vanilla (S)GD while using less memory on rather toy datasets with high-resolution images. However, concerns about the lack of comparison with other memory-efficient training methods, such as checkpointing, and the absence of training time metrics were initially raised by reviewers. These issues were addressed in the rebuttal phase, where the authors provided additional clarification and experiments, making the evidence more robust.